# MoRIC: A Modular Region-based Implicit Codec for Image Compression

**Gen Li**[*], **Haotian Wu**[*][†], **Deniz Gündüz**
Department of Electrical and Electronic Engineering
Imperial College London, London SW7 2AZ, U.K.
{gen.li22, haotian.wu17, d.gunduz}@imperial.ac.uk
**https://eedavidwu.github.io/MoRIC/**

## Abstract

We introduce Modular Region-Based Implicit Codec (MoRIC), a novel image compression algorithm that relies on implicit neural representations (INRs). Unlike previous INR-based codecs that model the entire image with a single neural network, MoRIC assigns dedicated models to distinct regions in the image, each tailored to its local distribution. This region-wise design enhances adaptation to local statistics and enables flexible, single-object compression with fine-grained rate-distortion (RD) control. MoRIC allows regions of arbitrary shapes, and provides the contour information for each region as separate information. In particular, it incorporates adaptive chain coding for lossy and lossless contour compression, and a shared global modulator that injects multi-scale global context into local overfitting processes in a coarse-to-fine manner. MoRIC achieves state-of-the-art performance in single-object compression with significantly lower decoding complexity than existing learned neural codecs, which results in a highly efficient compression approach for fixed-background scenarios, e.g., for surveillance cameras. It also sets a new benchmark among overfitted codecs for standard image compression. Additionally, MoRIC naturally supports semantically meaningful layered compression through selective region refinement, paving the way for scalable and flexible INR-based codecs.

## 1 Introduction

Traditional image and video codecs [5, 51, 9] rely on hand-crafted transforms to capture statistical correlations in the data, and exploit redundancies rooted in human perception for compression. Recently, deep learning-based codecs have demonstrated remarkable rate-distortion (RD) performance by replacing manually designed analysis and synthesis transforms with neural networks [2, 15, 22]. However, this advancement comes with significantly increased model complexity and decoding cost that often far exceed those of commercial codecs, limiting their practicality in real-world applications [26, 54]. Furthermore, these neural approaches typically require large-scale training or advanced models [47] to generalize across diverse data distributions. Consequently, developing codecs that balance low computational complexity and high performance remains a critical open challenge [58].

To address these limitations, overfitted codecs have emerged, leveraging implicit neural representations (INRs) [50] to compress each image individually with low decoding complexity. INRs employ lightweight multi-layer perceptrons (MLPs) to parameterize continuous functions [48, 37], with the quantized network weights serving as a compact, signal-specific compressor [48, 37]. First INR-based

---

[*]These authors contributed equally to this work and are listed alphabetically.
[†]Corresponding author: Haotian Wu (haotian.wu17@imperial.ac.uk).

39th Conference on Neural Information Processing Systems (NeurIPS 2025).

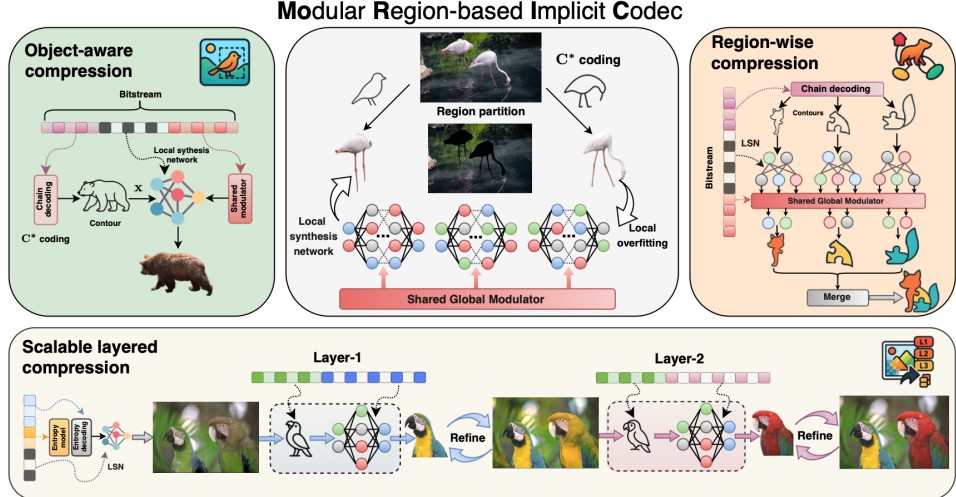

Figure 1: Illustration of the proposed framework, highlighting its key features: precise single-object compression, region-wise adaptive coding, and scalable layered compression. MoRIC enables fine-grained control over regions and objects for overfitted codecs, improving RD performance through region-wise fitting and global-local modulation. It also introduces layered compression, supporting semantically meaningful coarse reconstructions to be progressively refined with additional bits.

image codec, COIN, was introduced in [16]. It was later followed by the COMBINER series [21, 23], the COOL-CHIC series [30, 31], and the C3 series [27, 3], which have gradually improved the RD performance and perceptual quality. Notably, these approaches now surpass classical codecs such as BPG [5] and HEVC [51], while maintaining low decoding complexity.

Despite these advances, overfitted codecs still lag behind classical and neural codecs like VTM [10] and MLIC$^{++}$ [25, 26] in RD performance. Achieving higher fidelity often demands more complex networks, which, in turn, increases the bitrate. Architectural innovations for INRs in the context of data compression remain underexplored, with most overfitted codecs employing fixed architectures across diverse images, limiting both performance and adaptability. In practice, images exhibit varying statistical characteristics and compression difficulty, even across different regions within the same image. These limitations underscore the need for a new overfitted codec framework that improves performance while enabling flexibility and modularity.

Unlike previous codecs that overfit a single neural network to the entire image, we propose a "divide and conquer" approach, called MoRIC. MoRIC first partitions the image into distinct regions, and overfits a separate local synthesis network (LSN) to each region. This allows specialization to local distributions, and improves compression efficiency by reducing the complexity of individual LSNs. Unlike transform-based codecs, INRs are not limited to fixed rectangular shapes, and can be trained for regions of arbitrary shapes. However, for reconstruction of the whole image, we need to transmit the contour information of each region (except the background) separately. We note that a single INR trained for the whole image implicitly learns not only the statistical characteristics of all the image regions, but also the contour information. We, instead, employ a dedicated efficient contour coding scheme called C$^{*}$. MoRIC further employs a shared global modulator that injects global context to mitigate artifacts from local-only overfitting. Together, these components enable fine-grained control of the RD trade-off and introduce greater flexibility and representational capacity to the codec.

Our design is motivated by three key findings: (1) **Fine-grained and flexible region-based overfitting** improves RD performance and adaptability. (2) **Chain coding** enables efficient contour compression (lossy and lossless), and supports modular INRs with faster convergence via cross-region modulation. (3) **Semantic generalization** emerges from combining global modulation with local overfitting, allowing flexible layered compression with semantic-aware synthesis beyond pixel-level fidelity.

MoRIC enables modular region-based compression with low decoding complexity, achieving *state-of-the-art* performance among overfitted codecs across diverse scenarios. It accurately overfits target regions while reducing contour overhead, *outperforming the leading neural codec MLIC$^{++}$*

*with decoding complexity below* 800 *MACs/pixel* on single-object compression tasks. Its multiscale and coarse-to-fine modulation enhances reconstruction quality and streamlines coding through dynamic selection of regions requiring finer detail, enabling fine-grained complexity control and faster training. As a result, *MoRIC establishes new benchmarks for overfitted codecs*, surpassing VTM in standard image compression tasks. Moreover, *MoRIC is the first overfitted codec to support layered compression*, enabling semantic-aware, progressive reconstruction across regions. This design balances distortion, computation, and compression cost, paving the way for future research on scalable, flexible overfitted codecs. Our main contributions are:

- We propose MoRIC, a modular region-based implicit codec that overfits images region by region. This region-wise compression design improves adaptation to local content distributions and supports flexible, region-specific control for enhanced compression efficiency.

- We introduce adaptive chain coding, named $C^*$ coding, for efficient lossy and lossless contour coding, combined with shared modulation in a coarse-to-fine fashion, enabling MoRIC to achieve state-of-the-art single-object compression with significantly reduced decoding complexity. Notably, the open-sourced $C^*$ coding can be of independent interest for broader INR-based research beyond compression.

- Leveraging region-based compression, enhanced by $C^*$ coding and a global-local modulation mechanism, MoRIC sets up new baselines among overfitted codecs for standard full image compression task while enabling fine-grained RD control and region-specific adaptation.

- Moreover, MoRIC reveals that region-specific codecs can generalize across regions, enabling semantically meaningful reconstructions. Its layered, coarse-to-fine design allows local codecs to infer and progressively refine other regions, with additional models selectively transmitted to enhance fidelity, supporting scalable, flexible layered compression.

## 2 Related work

### 2.1 Overfitted codec

Overfitting a parametric function to a single image offers a low-complexity alternative to standard neural codecs. COIN [16] pioneered this by training a lightweight INR to map pixel coordinates to RGB values, with quantized parameters forming the bitstream. Later works [45, 46, 17] employed meta-learning for improved generalization. However, the COIN series still lagged behind standard autoencoder (AE) based codecs in RD performance. To address this, the COMBINER series [21, 23] introduces variational INRs and leverages relative entropy coding for RD-optimized compression.

The COOL-CHIC series [30] further enhances RD performance by incorporating learnable latent and entropy models. Its extensions, including COOL-CHICv2 [31], C3 [27, 3], and COOL-CHICv3/v4 [7, 32], introduce significant architectural advances such as soft-rounding, Kumaraswamy noise, conditional entropy coding, and Wasserstein distortion. These innovations enable performance that surpasses classical codecs like BPG [5] and HEVC [51]. More recently, LotteryCodec [56] is proposed to encode image statistics into the random network substructure for flexible complexity and better performance. Despite this progress, current overfitted codecs still fall short of advanced codecs like VTM [10] or MLIC$^{++}$ [25] in RD performance. Moreover, they lack support for diverse needs such as layered compression or fine-grained regional control. These limitations motivate our development of a more sophisticated, low-cost overfitted codec with flexible coding strategies.

### 2.2 Region partitioning for INRs

By modeling signals as continuous functions, INRs offer expressive representations for various tasks such as view synthesis, image inpainting, and super-resolution [50, 37, 13]. Partitioning the input space into local regions has proven effective in improving detail fidelity for INR-based models [24, 53]. Chabra et al. [11] introduced local latent vectors per partition, later extended by Mehta et al. [36] using a modulator network. Follow-up work explored multi-resolution partitioning [14, 55] and hierarchical tree structures for parameter sharing [57]. Input region partitioning can also accelerate training for faster convergence by enabling coarse-to-fine modeling [43], semantic segmentation [35], and input decoupling [34]. Ashkenazi et al. [1] proposed croppable INRs that support editability without retraining and fine-grained downstream control.

Although shape-based image/video coding has received significant research interest in the past [44], it has not found applications in practice. In transform-based coding, this requires designing specific transforms [49, 33] for arbitrary shapes, which creates significant complexity. On the other hand, INRs are particularly appropriate for this as they can be trained on arbitrary shapes.

## 3 Methodology

### 3.1 Preliminary

**Standard overfitted codec.** At its core, an overfitted codec compresses each sample by fitting it to a neural function, with quantized and entropy-coded parameters forming the bit cost. As shown in Fig. 2 (a), a standard overfitted codec models a data point $S$ using a function $g_W$ and a latent vector $z$. This fitting process is indicated by $\mathcal{F}$, while $Q$ and $\mathcal{A}$ represent quantization and entropy coding. At decoding, $S$ is reconstructed using the quantized $g_{\hat{W}}$ and $\hat{z}$ as $\hat{S} = g_{\hat{W}}(\hat{z})$, where hat notation denotes quantized variables. This results in an average distortion $D = \mathbb{E}_{S \sim p_s}\big[d(S, g_{\hat{W}}(\hat{z}))\big]$ at a rate

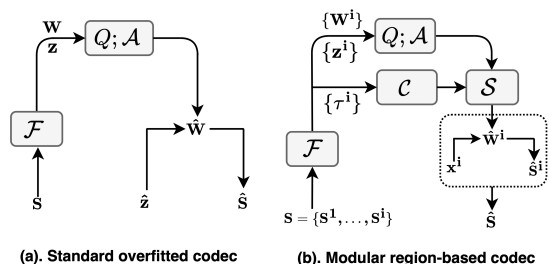

(a). Standard overfitted codec     (b). Modular region-based codec

Figure 2: From standard overfitted codec to MoRIC.

$R = \mathbb{E}_{S \sim p_s}\Big[-\log_2 p_{\hat{\psi}}(\hat{z}) - \log_2 p(\hat{W}) + R_{\hat{\psi}}\Big]$, where $d$ denotes a distortion metric and $p_{\hat{\psi}}(\cdot)$ is the estimated distribution of $\hat{z}$ using an estimation model $\hat{\psi}$. In practice, entropy coding of $\hat{z}$ relies on a lightweight auto-regressive entropy model (ARM) [30] for distribution estimation, introducing additional cost $R_{\hat{\psi}}$ for transmitting its parameters.

**Modular layered codec.** Taking overfitting to a more fine-grained level, we propose a modular region-based codec (MoRIC) that assigns dedicated INRs to distinct regions of a data sample (see Fig. 2 (b)). Specifically, MoRIC first partitions the input image $S$ into $N$ mutually exclusive regions $\{S^1, \ldots, S^N\}$, where the position of region $S^i$ within the image is denoted by $x^i$. Each $S^i$ is fitted using a distinct neural function $g_{W^i}$ and a latent vector $z^i$, both of which are quantized and entropy-coded. The region contours $\tau^i$ are compressed using a chain coding scheme $\mathcal{C}(\cdot)$. During decoding, $\tau^i$ guide the selection of parameter sets $\{W^i, z^i\}$ via a selection function $\mathcal{S}(\cdot)$, enabling the reconstruction region-by-region as $\hat{S} = \bigcup_{i=1}^{N} g_{W^i}(\hat{z}^i, x^i)$, where $\bigcup$ denotes the region merging operation. The expected distortion is $D = \mathbb{E}_{S \sim p_s}\Big[d(S, \hat{S})\Big]$, and the rate is

$R = \mathbb{E}_{S \sim p_s}\Big[\sum_{i=0}^{N}\big[-\log_2 p_{\hat{\psi}}(\hat{z}^i) - \log_2 p(\hat{W}^i) - \log_2 p(\tau^i)\big] + R_{\hat{\psi}}\Big]$. While each region uses different $W^i$, a shared ARM is employed to improve compression efficiency.

### 3.2 Adaptive contour compression with C* coding

Inspired by [12], we propose an adaptive chain coding method (called C* coding) for efficient contour compression in region-based overfitted codecs. It ensures (a) semantic coverage of target objects against artifacts, (b) robustness to high-curvature via adaptive dilation, and (c) a balanced RD trade-off. As illustrated in Fig. 3, C* coding comprises four key modules: angle matching, global resetting, offset adaptation, and contour dilation.

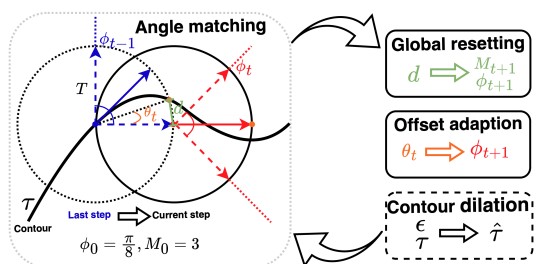

Figure 3: Illustration of C* coding.

**Lossy contour compression.** Starting from an initial point with step length $T$ (in pixels) and initial angle $\phi_0$, the direction space is evenly

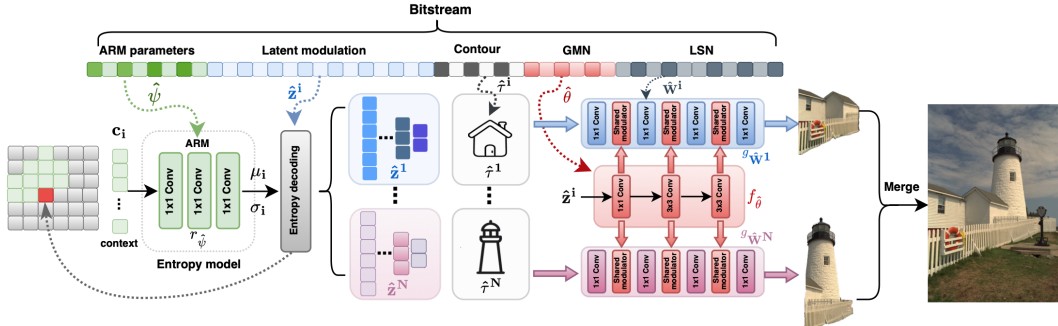

Figure 4: Workflow of MoRIC: the image is partitioned into $N$ regions, each compressed using a distinct LSN $g_{\hat{\boldsymbol{W}}^i}$ modulated with a shared GMN $f_{\hat{\boldsymbol{\theta}}}$.

divided into $M_0$ angles for the next step. At each step, we compute the offset angle $\theta_t$ between the nearest angle candidate $\frac{\phi_t}{M_t}$ and the contour point located $T$-distance ahead, along with its distance offset $d$. Based on $d$, a global reset is applied to update $M_{t+1}$ and $\phi_{t+1}$ if the previous offset adjustment is considered too large. Subsequently, $\phi_{t+1}$ is adaptively adjusted according to $\theta_t$ for better angle-matching in the next step. After each coding round, a dilation is selectively applied if the target region is not well-covered by the contour, based on an error threshold $\epsilon$, followed by re-coding. This iterative process produces the final lossy contour $\hat{\tau}$. The bit cost per step is $\lceil \log_2(M_t) \rceil + 1$, accounting for angle selection and reset signaling. A move-to-front (MTF) transform [18, 6] is further applied to reduce redundancy. See Algorithm 1 and Fig. 12 in Appendix D.2 for details.

**Lossless contour compression.** For lossless coding, we fix $\phi = 2\pi$ and $M = 8$, followed by an MTF transform. To further reduce bit cost, we additionally incorporate three alternative strategies: (1) shared vertex descriptions via vertex chain coding (VCC) [8], (2) directional encoding based on prior directions using three orthogonal chain codes (3OT) [42], and (3) angular difference encoding through normalized angle difference chain coding (NAD) [59]. All schemes will be open-sourced; see Appendix D.2 and supplementary materials for more details.

### 3.3 MoRIC architecture

As shown in Fig. 4, the MoRIC architecture comprises four key components: **local synthesis networks** (LSNs), a **global modulation network** (GMN), **latent modulations** ($\boldsymbol{z}^i$), and a shared **ARM** ($r_{\boldsymbol{\psi}}$). Specifically, each LSN ($g_{\boldsymbol{W}^i}$) is optimized for a designated region to map local coordinates to RGB values. The GMN ($f_{\boldsymbol{\theta}}$) is globally optimized across all regions to modulate each LSN layer by layer. Latent modulations $\boldsymbol{z}^i$ are learnable vectors, and ARM is a shared entropy model used to compress all $\boldsymbol{z}^i$. More architectural and algorithmic details are provided in Appendix D.

**Local synthesis networks (LSNs).** For each region, the corresponding LSN takes the coordinates within the contour ($\boldsymbol{x}^i$) as input and reconstructs their pixel intensities. As shown in Fig. 5, each $g_{\boldsymbol{W}^i}$ consists of $L_s = 4$ convolutional layers, each followed by a shared modulator $\mathcal{M}(\cdot)$. At each layer, $\mathcal{M}(\cdot)$ concatenates the output of the previous layer with a modulation vector provided by $f_{\boldsymbol{\theta}}$, and passes it through a shared-weight MLP to generate the next-layer input.

A key innovation here is that the parameters of $\mathcal{M}(\cdot)$ are shared between regions, which offers: (a) reduced bitrate via parameter sharing, (b) coarse-to-fine multiscale modulation across layers, and (c) implicit fusion of local and global features. This design simplifies local coding while injecting global context, boosting compression efficiency and RD performance.

**Latent modulation.** For the $i$-th region, latent modulation is defined as $\boldsymbol{z}^i \triangleq \{\boldsymbol{z}_1^i, \boldsymbol{z}_2^i, \ldots, \boldsymbol{z}_L^i\}$, where each $\boldsymbol{z}_j^i \in \mathbb{R}^{\frac{K^i}{4^{j-1}}}$ is a learnable vector with the same spatial shape as the region $i$, corresponding to a $2^{j-1} \times 2^{j-1}$ downsampling of its coordinates, with $K^i$ being the pixel count of region $i$.

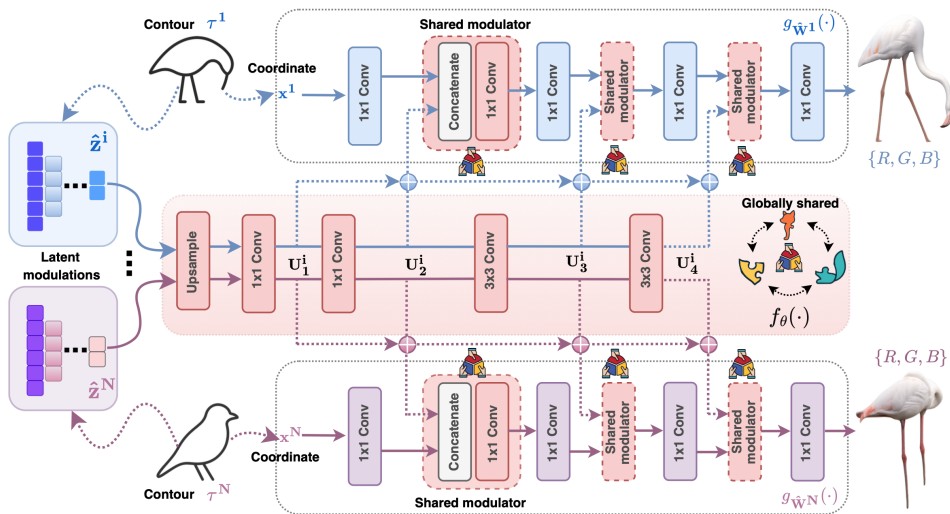

Figure 5: Illustration of the proposed PCM mechanism. Red blocks with shared icons indicate parameter-shared components across regions. Each LSN is modulated using multi-scale features generated by $f_{\hat{\theta}}$ from the corresponding latent vector $\hat{z}^i$.

**Global modulation network (GMN).** As illustrated in Fig. 5, GMN $f_{\theta}$ consists of an upsampling module followed by $L_m = 4$ convolutional layers. It takes the quantized $\hat{z}^i$ as input and generates hierarchical modulation vectors for each LSN. Specifically, $\hat{z}^i$ is first upsampled via transposed convolutions to form $\boldsymbol{U_0^i} = [\boldsymbol{u_1^i}; \boldsymbol{u_2^i}; \ldots; \boldsymbol{u_L^i}] \in \mathbb{Z}^{L \times K^i}$, with each $\boldsymbol{u_j^i} \triangleq Upsample(\hat{z}_j^i) \in \mathbb{Z}^{K^i}$. The resulting $\boldsymbol{U_0^i}$ is then processed through successive convolutional layers as: $\boldsymbol{U_k^i} = f_{\theta}^{(k)}(\boldsymbol{U_{k-1}^i})$, where $f_{\theta}^{(k)}$ and $\boldsymbol{U_k^i}$ denote the operation and output at the $k$-th layer of $f_{\theta}$ for the $i$-th region, respectively.

**Progressive Concatenated Modulation (PCM).** To inject global context into local synthesis in a coarse-to-fine manner, we propose the PCM mechanism. PCM progressively concatenates latent-derived features to enable multiscale modulation across regions. For the $k$-th layer of $g_{\boldsymbol{W}^i}$, the modulation vector is defined as $\boldsymbol{M_k^i} = \text{Concatenate}(\boldsymbol{U_1^i}, \ldots, \boldsymbol{U_k^i})$. At each LSN layer, modulation operation $\mathcal{M}(\cdot)$ concatenates $\boldsymbol{M_k^i}$ with the current layer output and processes it through a shared-weight MLP to produce the input for the next layer. Intuitively, this design consistently injects global context and allows each region to dynamically refine its representation. Notably, MoRIC with PCM serves as a general framework that can support alternative modulation strategies, such as [40, 36].

### 3.4 RD optimization

To balance the RD trade-off, MoRIC with an $N$-region configuration is trained to fit each data point into the parameter set: $\boldsymbol{\Omega} \triangleq \left\{ \psi, \boldsymbol{\theta}, \bigcup_{i=1}^{N} \left( \boldsymbol{z}^i, \boldsymbol{\tau}^i, \boldsymbol{W}^i \right) \right\}$, by minimizing the following loss:

$\mathcal{L}(\boldsymbol{\Omega}) = \sum_{i=0}^{N} \left[ d(\boldsymbol{S^i}, \hat{\boldsymbol{S}}^i) + \lambda R(\hat{\boldsymbol{z}}^i) \right]$, where $\hat{\boldsymbol{S}}^i = g_{\boldsymbol{W}^i}(f_{\theta}(\hat{\boldsymbol{z}}^i), \boldsymbol{x}^i)$ denotes the reconstruction of the $i$-th region, $R(\hat{\boldsymbol{z}}^i)$ is the rate term of $\hat{\boldsymbol{z}}^i$, and $\lambda$ is a hyperparameter that controls the RD trade-off.

During training, latent modulation $\boldsymbol{z}^i$ undergoes soft-rounding with Kumaraswamy noise, while hard-rounding is used at inference. The loss excludes rate terms for $\boldsymbol{\tau}^i$ and $\{\boldsymbol{\theta}, \boldsymbol{W}^i, \psi\}$, due to their minor contributions. In practice, $\boldsymbol{\tau}^i$ is encoded using C* coding, and network parameters are quantized and entropy-coded via a non-learned distribution. All components are included in the final bitstream. Implementation details are provided in Appendix D.4.

### 3.5 Advantages of MoRIC

**Single-object compression.** MoRIC can be especially promising for fixed-background (static camera) scenarios like surveillance systems, where detected objects of arbitrary shapes are compressed individually, while the fixed background can be assumed to be known by the receiver.

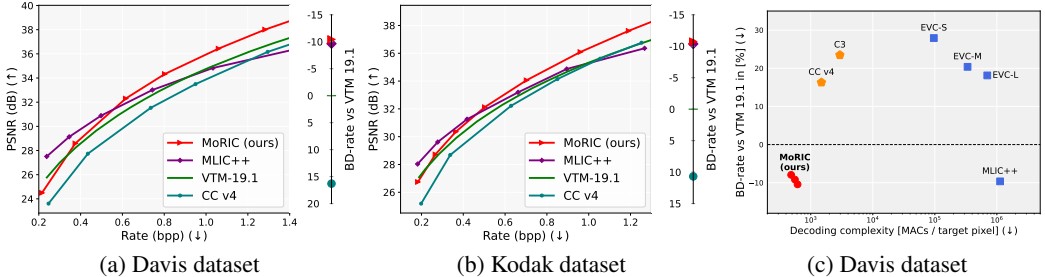

| (a) Davis dataset | (b) Kodak dataset | (c) Davis dataset |
|---|---|---|

Figure 6: RD performance for single-object compression. (a-b) RD curve and BD rate on Davis and Kodak datasets, respectively. (c) BD-rate vs. decoding complexity on Davis dataset.

Table 1: Effect of $C^*$ coding for contour compression across various granularity levels.

| Step length | IoU (%) ↓ | bpp of $\tau$ ↓ | RD performance | |
|---|---|---|---|---|
| | | | Total bpp ↓ | PSNR (dB) ↑ |
| 1 | 100% | 0.0637 | 0.213 | 24.76 |
| | | | **0.580** | **32.16** |
| 4 | 93.66% | 0.0409 | 0.204 | 24.56 |
| | | | 0.621 | 32.01 |
| 6 | 91.07% | 0.0244 | 0.192 | 24.46 |
| | | | 0.611 | 31.90 |
| 10 | 87.31% | 0.0140 | **0.186** | **24.43** |
| | | | 0.621 | 31.80 |

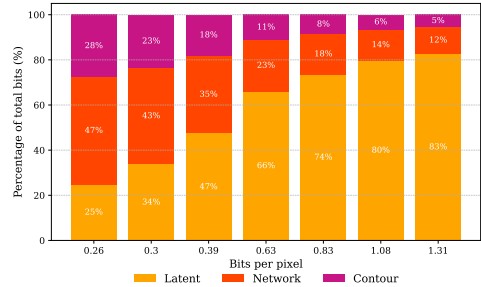

Figure 7: Compression overhead breakdown of MoRIC for single-object compression.

**Region-wise compression.** MoRIC establishes a new paradigm for overfitted codecs through a modular, region-wise design that improves the RD performance, enables flexible complexity control across regions, and offers potential for accelerated encoding (similarly to [43, 35, 34, 1]).

**Scalable layered compression.** MoRIC reveals that overfitted codecs can achieve semantic generalizability: its shared PCM and ARM enable layered, coarse-to-fine compression, allowing local INRs to progressively infer and refine other regions via region-wise transmission.

# 4 Experimental results

We evaluate MoRIC on DAVIS [39], Kodak [28] and CLIC2020 datasets [52], comparing it with classical codecs (VTM [9]), autoencoder-based neural codecs (EVC [54], MLIC$^{++}$[26]), and recent overfitted codecs (C3[27], COOL-CHIC v4 [32], and LotteryCodec[56]). RD performance is assessed using peak signal-to-noise ratio (PSNR) on RGB channels and BD-rate [20], while MACs per pixel and coding latency are reported to evaluate decoding complexity and coding efficiency. Region masks are directly taken from DAVIS and generated for Kodak using SAM2 [41]. Additional experiments, datasets, and visualizations are provided in the Appendix and supplementary materials. We begin with the single-object compression task, which focuses on precise compression of targeted regions or objects (experimental setup detailed in Appendix E.2). Next, we evaluate MoRIC on the standard full image compression task. Finally, we explore a layered compression scenario, where the source is progressively encoded and reconstructed in a coarse-to-fine manner.

## 4.1 Single-object compression

**RD performance.** As shown in Figs. 6a and 6b, MoRIC achieves state-of-the-art (SOTA) RD performance on DAVIS and Kodak for single-object compression, outperforming MLIC$^{++}$ (current leading neural codec) with BD-rate improvements of $-10.43\%$ and $-10.73\%$ over VTM-19.1. By overfitting only target pixels, it ensures accurate reconstruction with reduced computation. Although less competitive at very low bitrates due to contour and network overhead, it still outperforms other

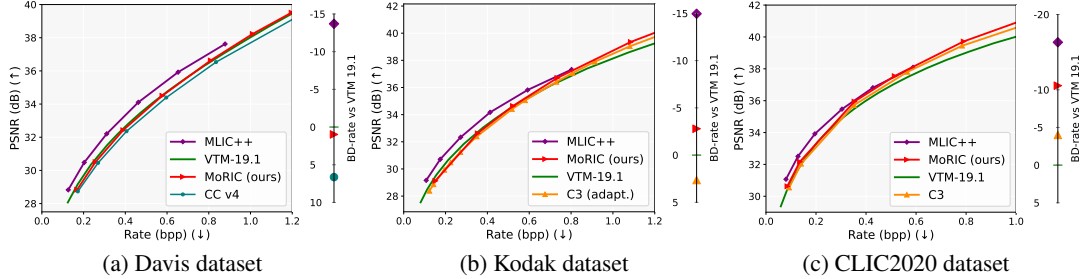

(a) Davis dataset     (b) Kodak dataset     (c) CLIC2020 dataset

Figure 8: RD performance with BD-rate results for standard image compression on (a) Davis, (b) Kodak, and (c) CLIC2020 datasets.

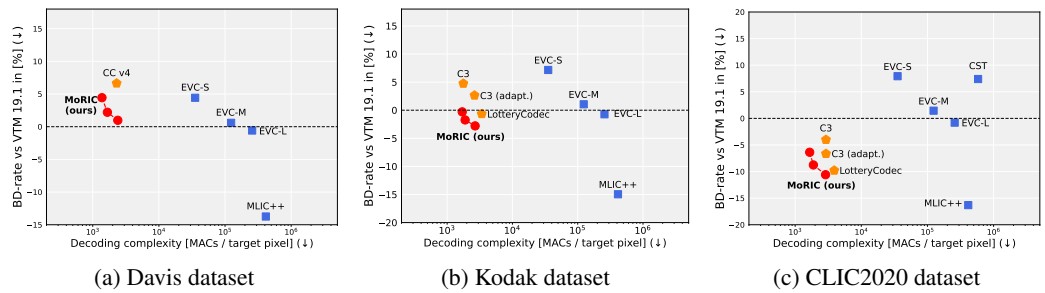

(a) Davis dataset     (b) Kodak dataset     (c) CLIC2020 dataset

Figure 9: Trade-off between RD performance and decoding complexity on (a) Davis. (b) Kodak, and (c) CLIC2020 datasets.

overfitted codecs thanks to its shared-layer design. Additional results and ablations in Figs.15 and 16 in the Appendix further validate the effectiveness of our design.

**Decoding efficiency.** MoRIC offers low decoding complexity in single-object compression, achieving better RD performance with fewer than 800 MACs/pixel, far lower than all AE-based codecs (see Fig. 6c). A detailed analysis of decoding complexity, latency, and BD-rate is provided in Fig.17 and Table 5 in the Appendix. These results underscore the efficiency of region-based compression by overfitting only target pixels for accurate and efficient reconstruction.

**Effect of contour compression.** To evaluate the impact of $C^*$ coding, Table 1 reports intersection-over-union (IoU), contour bit cost, and RD performance. As step length increases, the bit cost drops, with RD performance remaining acceptable at low bpp values, but degrading sharply at high bpp, highlighting the benefit of lossy contour coding in low-bpp regimes and the importance of contour accuracy at high bpp. Fig. 7 further breaks down compression overhead across various bitrates, showing that contour bits dominate at low bpp, but are negligible at high bitrates, validating the efficiency of our approach. More details and visualization of $C^*$ coding are provided in the Appendix.

## 4.2 Full image compression

**RD performance.** As shown in Figs. 8a, 8b and 8c, MoRIC achieves state-of-the-art RD performance among overfitted codecs for the standard full image compression task (as opposed to compressing only the region/object of interest considered above). It achieves a $-5.0\%$ BD-rate reduction over C3 on Kodak, $-5.51\%$ over COOL-CHIC (v4) on DAVIS, and $-5.48\%$ over C3 on CLIC2020. Notably, MoRIC surpasses VTM-19.1 on the Kodak and CLIC2020 datasets with BD-rate reductions of $-2.79\%$ and $-10.58\%$, respectively, establishing a new benchmark among overfitted codecs. In comparison, the previous state-of-the-art LotteryCodec achieved $-0.66\%$ and $-9.79\%$. Like other overfitted codecs, MoRIC still falls short of leading neural codecs MLIC++ in terms of the RD performance, but achieves over 100× lower decoding complexity. Additional results and ablations in the Appendix (Figs.20-21 and Tables. 6-12) further validate the effectiveness of

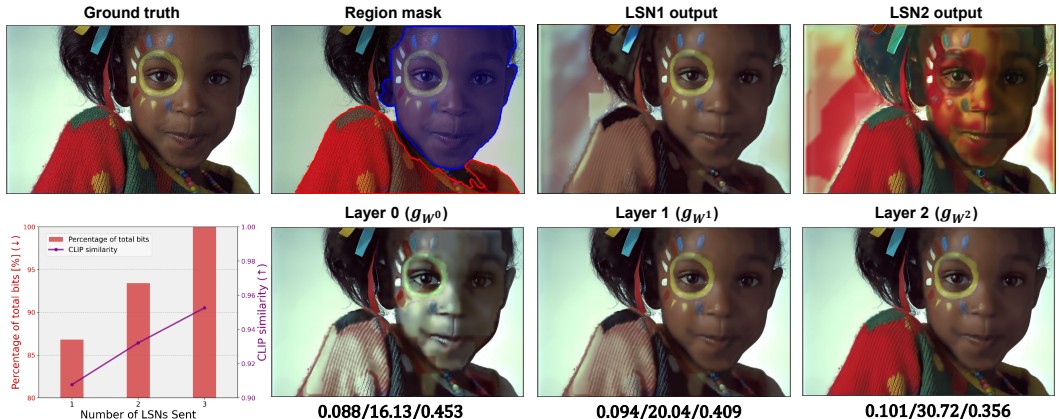

| Ground truth | Region mask | LSN1 output | LSN2 output |
| --- | --- | --- | --- |

| | Layer 0 ($g_{W^0}$) | Layer 1 ($g_{W^1}$) | Layer 2 ($g_{W^2}$) |
| --- | --- | --- | --- |
| | 0.088/16.13/0.453 | 0.094/20.04/0.409 | 0.101/30.72/0.356 |

Figure 11: Layered compression with MoRIC: The background LSN $g_{W^0}$ is transmitted first (layer 0), followed by $g_{W^1}$ (blue region, layer 1) and $g_{W^2}$ (red region, layer 2). The first row shows outputs from each individual LSN; the second row shows the cumulative reconstructions after each layer. Bitrate in bits-per-pixel (bpp)/ PSNR (dB)/ LPIPS scores are reported below each image.

our design. Interestingly, lossy contour compression reduces the bitrate with minimal impact on distortion, thanks to global-local modulation design across regions.

**Effect of number of regions.** We conduct experiments under different region configurations (over 10 Kodak images). As shown in Fig. 10, increasing the number of regions improves the RD performance, with the largest gain from 1 to 2 regions. Since the gains are diminishing with $N$ while the complexity increases, we consider up to 3 regions for strong performance with manageable overhead. Detailed results are provided in Fig. 24 with visualization in Fig. 25 in the Appendix.

**Decoding efficiency.** MoRIC achieves strong RD performance with low decoding complexity (Fig. 9). Compared to other overfitted codecs, it achieves superior RD performance with fewer decoding operations. *To the best of our knowledge, MoRIC is the first neural codec to surpass VTM on Kodak in RD performance ($-2.79\%$ BD-rate) while maintaining low decoding complexity (below $2000$ MACs/pixel).* Additional results on decoding complexity, coding latency, ablation study, and BD-rate across different datasets are provided in Fig.9 and Table 6, 10 12.

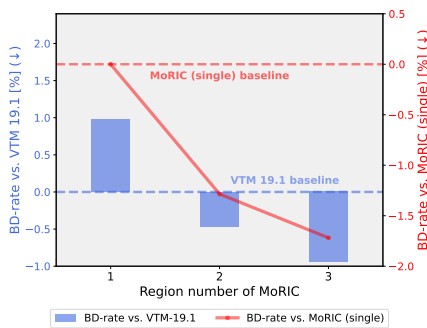

Figure 10: Effect of region numbers.

### 4.3 Scalable layered compression

As shown in Figs.1 and 11, MoRIC naturally supports layered, coarse-to-fine compression by first transmitting the background LSN to produce a coarse reconstruction, followed by region-specific LSNs for progressive refinement. Notably, even with only the background LSN, object structure and semantics remain clearly recognizable despite color inaccuracies, highlighting the ability of MoRIC to generalize between regions. As more bits are allocated for additional LSNs with region-specific refinement, color and texture details improve progressively.

More examples in Figs. 28 and 29 reveal that common features like shared colors are captured by the global modulator, enabling local LSNs to infer them correctly. In contrast, region-specific features and colors require dedicated INRs for precise refinement. From Fig. 30, we observe that cross-region inference is more challenging in the low-bpp regime, while richer latent modulations at higher bpp enable more accurate context and feature reconstruction. This layered compression capability of MoRIC is especially beneficial for high-resolution images with many LSNs, enabling progressive refinement and efficient and controllable region-wise reconstruction.

# 5 Conclusion

We presented MoRIC, a modular region-based implicit codec that compresses images via region-wise overfitting. By assigning dedicated networks to distinct regions, MoRIC enables fine-grained RD control and precise single-object compression. Our approach integrates adaptive chain coding for efficient contour compression and employs GMN for coarse-to-fine multiscale modulation of local synthesis. MoRIC achieves the SOTA performance in both single-object and full image compression tasks with low decoding complexity. Furthermore, it naturally introduces layered compression capabilities resulting in more scalable and flexible overfitted codecs for real-world applications.

**Limitations.** While the region-based design of MoRIC helps with convergence, its high encoding complexity remains a challenge. Therefore, like other overfitted codecs, MoRIC is especially beneficial for multi-user streaming applications, supporting one-time encoding and repeated decoding. Potential acceleration strategies include meta-learning and region-wise mixed-precision training.

**Future work.** MoRIC can be extended to video coding by leveraging contour correlations and sharing modulations across adjacent groups of frames (GoFs), improving the RD performance and convergence speed. While it currently uses SAM2 for region selection, future work can integrate segmentation into the coding process to explore trade-offs between granularity and performance.

## Acknowledgments and Disclosure of Funding

We acknowledge funding from the UKRI for the projects AI-R (ERC Consolidator Grant, EP/X030806/1) and INFORMED-AI (EP/Y028732/1).

## Author contribution

D.G. conceived and supervised the project, and developed its overall vision. H.W. implemented the initial codebase and designed the experiments. G.L. designed the chain-coding, local-global mechanism, and optimized performance. H.W. and D.G. wrote and revised the paper.
These authors contributed equally and are listed alphabetically: G. Li and H. Wu.

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

## Appendix Contents

## A    Open resources

**(a) Datasets.**    We open-source our region-based coding dataset, including the DAVIS set (18 images) used in our experiments and our region segmentation results for the Kodak (24 images) and CLIC (41 images) datasets. Additionally, we provide a new fixed-background dataset to support future research on region-based compression, particularly in fixed-background scenarios.

**(b) Adaptive chain coding method.**    We open-source both lossy and lossless variants of $C^*$ to facilitate future research and development for modular region-based codec (which is included in the supplementary materials and to be released publicly after review). **Notably, $C^*$ coding may be of independent interest for broader INR-based research beyond compression.**

**(c) Code and checkpoints.**    Code and checkpoints are released in the project page.

## B    Broader impacts

This work presents a technical contribution in image compression with no immediate societal impact. Nonetheless, potential downstream effects may arise depending on application context.

**Positive impacts**    (1). Efficiency: Its ultra-low decoding complexity supports energy-efficient, scalable deployment in bandwidth-constrained settings like IoT and mobile devices, especially in multi-user streaming scenarios, significantly reducing the energy footprint of edge devices in the future. (2). Privacy: Region-based compression enables selective transmission and reconstruction, supporting privacy-preserving use cases (e.g., face masking). (3). Semantic-aware content delivery: Layered, region-based decoding benefits adaptive streaming and semantic-level rendering in AR/VR.

**Potential negative impacts**    However, region-based compression could also be misused to selectively manipulate or obscure image content. We encourage future work to integrate safeguards such as integrity verification and explainable compression decisions. All datasets used are publicly available, and we release our models and code to promote transparency and reproducibility.

## C    Baseline implementation

We evaluate our method on Davis and Kodak, where Davis is a standard benchmark featuring salient object regions, and Kodak is a standard dataset for image compression task. A detailed descriptions of the data used (including their masks) are provided in supplementary materials. MoRIC is compared against classical codec, VTM [9], as well as autoencoder-based neural codecs: EVC [54] (optimized RD performance with low decoding complexity), and MLIC++ [25] (state-of-the-art neural codec). Additionally, we compare with overfitted INR-based codecs such as C3 [27] (state-of-the-art overfitted image codec) and COOL-CHICv4 [32] (an optimized version of COOL-CHIC version with competitive performance from C3), and the most advanced LotteryCodec[56]. We measure PSNR on RGB channels and quantify RD performance using the BD-rate metric [20]. The baseline results were obtained using their official implementations or directly using the reported results from their papers. For VTM, we use the CompressAI library [4] to implement VTM-19.1 (YUV 10 bits), with additional details (code and datapoints) provided in the supplementary materials.

# D MoRIC implementation

## D.1 Region selection

For single-object compression, we directly use the provided region masks from the DAVIS dataset, while for the Kodak dataset, regions are manually selected using SAM2. The selection focuses on areas of contrast or saliency. We emphasize that the region selection in this work serves to investigate the potential of region-based compression. The masks are not optimized and can be further refined, with significant performance improvements expected. All masked datasets used in our experiments are provided in the supplementary material with open access to support future research on region-based compression.

## D.2 Adaptive region contour compression using $C^*$ coding

$C^*$ coding is tailored for region-based overfitted codecs. where we consider three main objectives:

(a) Semantic coverage: ensuring the full inclusion of the target object region to preserve semantic integrity and minimize object reconstruction artifacts. In typical region-based coding, the image is split into foreground and background. By covering the entire object, $C^*$ avoids boundary artifacts over objects, especially beneficial for the fixed-background scenarios. Background areas, which are generally less critical, can be approximated by the INR with minimal object-quality loss.

(b) Robustness to curvature: applying adaptive dilation to the coding process can simply prevent encoding failures, caused by high-curvature contours that may lead to ambiguous or looping paths. This simple yet effective mechanism enhances reliability in region-based coding.

(c) RD trade-off: balancing contour accuracy and compression cost is crucial in overfitted codec design. While lossy contours (without covering target region) may save bitrate by omitting parts of the object, they often lead to significant degradation in distortion performance.

Next, we detail the proposed $C^*$ coding scheme, as illustrated in Fig.12 and Algorithm1.

**Lossy compression.** At each step, we compute the offset angle $\theta_t$ between the current point and the contour point $T$ step-length ahead, i.e., the intersection with a radius-$T$ circle—and select the closest discrete angle $\frac{\phi_t}{M_t}$ as the next direction. Meanwhile, a offset distance $d$ is also computed.

Based on $d$, we determine whether the offset is too large, indicating the coding step has drifted from the contour. If so, we perform a global reset by setting the angle to $2\pi$ and increasing directional resolution. This is triggered when $d > 2T \sin\left(\frac{\phi_t}{4(M-1)}\right)$. After one adjustment step, $\phi$ and $M$ are reset to their initial values.

After a global reset, the angle is adapted based on the accumulated average offset $\bar{\theta}_t$. If $\bar{\theta}_t < \frac{\phi_t}{2(M-1)}$, the angle is halved, indicating the current direction is sufficiently accurate. If $\bar{\theta}_t > \frac{\phi_t}{M-1}$, the angle is doubled to allow finer directional correction, and $\bar{\theta}t$ is reset to zero. Otherwise, the angle remains unchanged: $\phi_{t+1} = \phi_t$.

After each coding round, a dilation step is selectively applied to ensure the contour fully encloses the target region. This prevents semantic loss and significant RD degradation caused by missing pixels outside the contour. Specifically, a $3 \times 3$ kernel is used to dilate the contour. A more detailed algorithm is presented in Algorithm 1.

**Lossless compression.** For the lossless compression approach, we refer to our supplementary materials, with detailed instructions and codes.

## D.3 Region-based latent configurations

Each $z^i$ consists of $L$ learnable multi-resolution vectors, defined as $z^i \triangleq \{z_1^i, z_2^i, \ldots, z_L^i\}$, where $z_j^i \in \mathbb{R}^{\frac{K^i}{4^{j-1}}}$ corresponds to the $j$-th resolution level for the $i$-th region, and $K^i$ denotes the number of pixels within that region. Specifically, $z_1^i$ matches the spatial resolution of the $i$-th region, and each subsequent $z_j^i$ is constructed by successively downsampling by a factor of $2 \times 2$, so that $z_L^i$ is

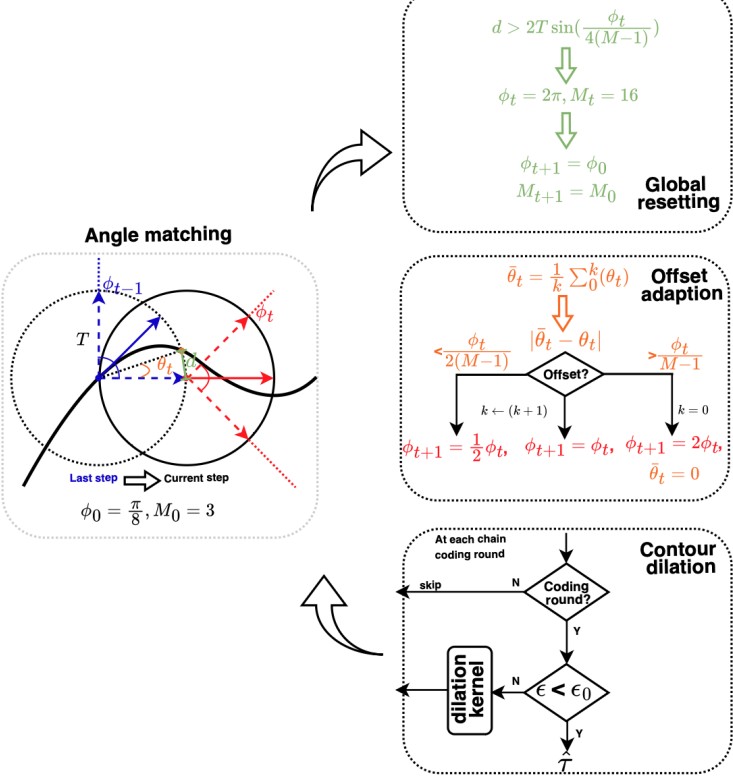

Figure 12: Illustration of the proposed $C^*$ coding, with details in each block.

reduced by $4^{L-1}$. In practice, all $z_j^i$ are initialized with the full image resolution $\mathbb{R}^{H \times W}$, where $H$ and $W$ are the original image height and width. A binary region mask is applied to isolate the area defined by the contour, which is then downsampled to construct each $z_j^i$. During training of each region, gradients are stopped for latent outside the region to ensure that only relevant networks and latent are updated. This allows efficient selection of latent features based on coordinates.

### D.4 Quantization and entropy coding details

In MoRIC, latent modulations $z^i$, and parameters of LSN for each region $W^i$, parameters of ARM and GMN go through quantization and entropy coding for compression, using standard methods, as in [27, 29]. More details can be seen in the Table. 2.

**Quantization-aware optimization.** Inspired by [27], we also adopt a two-stage quantization-aware optimization approach for each $z^i$. During training, $z$ is optimized in a continuous space with quantization approximated via Kumaraswamy noise, enabling differentiability. At inference, uniform quantization with hard-rounding is applied as:

$$\hat{z}^i = \begin{cases} \mathcal{S}_T(z^i) + u_{kum}, & \text{Training Stage I} \\ Q(z^i), & \text{Training Stage I \& Inference Stage,} \end{cases} \tag{1}$$

$\mathcal{S}_T$ is soft-rounding with temperature $T$, $Q$ is hard-rounding, and $u_{kum}$ is Kumaraswamy noise. The temperature and noise are annealed to shift from peaked to uniform distributions throughout training.

**Entropy coding.** Similar to prior works [2, 30, 38], we adopt a factorized auto-regressive model $r_\psi$ to estimate the distribution of each $\hat{z}^i$ for entropy coding. For each $i$-th region, every latent element of $\hat{z}_{k,j}^i$ (the $j$-th element of $z_k^i$) is conditioned on $C$ spatially neighboring elements $c_{k,j}^i \in \mathbb{Z}^C$ as:

$$p_\psi(\hat{z^i}) = \prod_{k,j} p_\psi(\hat{z}_{k,j}^i | c_{k,j}^i), \tag{2}$$

---
**Algorithm 1** C* coding of MoRIC
---
**Input:** Initial angle $\phi_0$, angle divisions $M_0$, and step length $T$, error threshold $\epsilon_0$,
initial $\hat{\tau} = \tau$, total iteration number $K$, accumulated step number $k = 0$, dilation kernel $\kappa(\cdot)$.
**Output:** Bitstream of contour $\boldsymbol{b}_\tau$.

1: **for** each iteration among the $K$ **do**
2:     **for** each $t$-th step until the coding of $\hat{\tau}$ is completed **do**
3:         $\theta_t, d \leftarrow \{T, \text{current position}\}$,                # Compute the offset angle and distance
4:         **if** $d > 2T \sin(\frac{\phi_t}{4(M_t-1)})$ **then**
5:             $\phi_t = 2\pi, M_t = 16$,                    # Need more accurate divisions
6:             $\phi_{t+1} = \phi_0, M_{t+1} = M_0$               # Reset angle and divisions
7:         **end if**                                  # Global resetting end
8:         $k = k+1, \bar{\theta}_t = \frac{1}{k} \sum_0^k (\theta_i)$,           # Update current average offset
9:         **if** $\bar{\theta}_t < \frac{\phi_t}{2(M_t-1)}$ **then**
10:            $\phi_{t+1} = \frac{1}{2}\phi_t$,                    # Offset is small, decrease the angle
11:         **else if** $\bar{\theta}_t > \frac{\phi_t}{(M_t-1)}$ **then**
12:            $\phi_{t+1} = 2\phi_t, \bar{\theta}_t = 0, k = 0$,       # Offset is large, increase the angle
13:         **else**
14:            $\phi_{t+1} = \phi_t$,
15:         **end if**                                 # Offset adaption end
16:     **end for**
17:     $\epsilon \leftarrow \{\tau, \hat{\tau}\}$,                       # Compute the error of object contour
18:     **if** $\epsilon > \epsilon_o$ **then**
19:         $\hat{\tau} \leftarrow \kappa(\hat{\tau})$,                        # Dilation the contour
20:     **else**
21:         Return $\hat{\tau}$ and break the loop         # Finish the contour coding
22:     **end if**
23: **end for**
24: $\boldsymbol{b}_\tau = \text{MTFT}(\boldsymbol{\tau})$                         # Move-to-front transformation
---

where $p_{\boldsymbol{\psi}}(\hat{\boldsymbol{z}}^{\boldsymbol{i}})$ is modeled by an integrated Laplace distribution. The expectation and scale parameters are estimated via context elements as $g \sim \mathcal{L}(\mu_{k,j}^i, \sigma_{k,j}^i)$, where $\mu_{k,j}^i, \sigma_{k,j}^i = r_{\boldsymbol{\psi}}(\boldsymbol{c}_{\boldsymbol{k,j}}^{\boldsymbol{i}})$. With this estimated distribution of $\hat{\boldsymbol{z}^i}$, a range coding algorithm is used (range-coder in PyPI) to compress $\hat{\boldsymbol{z}}^{\boldsymbol{i}}$. The resulting rate contributed by $\hat{\boldsymbol{z}}^{\boldsymbol{i}}$ is then given by:

$$R(\hat{\boldsymbol{z}}^{\boldsymbol{i}}) = -\log_2 p_{\boldsymbol{\psi}}(\hat{\boldsymbol{z}}^{\boldsymbol{i}}) = -\log_2 \prod_{k,j} p_{\boldsymbol{\psi}}(\hat{z}_{k,j}^i | \boldsymbol{c}_{\boldsymbol{k,j}}^{\boldsymbol{i}}) = \sum_{k,j} -\log_2 p_{\boldsymbol{\psi}}(\hat{z}_{k,j}^i | \boldsymbol{c}_{\boldsymbol{k,j}}^{\boldsymbol{i}}). \tag{3}$$

Then the total bit cost from the latent modulations can be expressed as $\boldsymbol{b_z} = \sum_{i=0}^{N} R(\hat{\boldsymbol{z}}^{\boldsymbol{i}})$.

**Parameter compression.** The parameters of neural networks: each LSN, GMN, and ARM are compressed with quantization operations and an unlearnable entropy model. Specifically, we first quantize $\boldsymbol{W}^{\boldsymbol{i}}$, $\boldsymbol{\theta}$, and $\boldsymbol{\psi}$ using a scalar quantizer $Q(\cdot, \Delta)$ with step size $\Delta$, defined as:

$$\hat{\boldsymbol{W}}^{\boldsymbol{i}} = Q(\boldsymbol{W}^{\boldsymbol{i}}, \Delta_{\boldsymbol{W}^i}), \hat{\boldsymbol{\theta}} = Q(\boldsymbol{\theta}, \Delta_{\boldsymbol{\theta}}), \text{and} \quad \hat{\boldsymbol{\psi}} = Q(\boldsymbol{\psi}, \Delta_{\boldsymbol{\psi}}). \tag{4}$$

The quantized parameters $\hat{\boldsymbol{W}}^{\boldsymbol{i}}$, $\hat{\boldsymbol{\theta}}$ and $\hat{\boldsymbol{\psi}}$ are then entropy-coded, where the discrete distribution of each quantized parameter is modeled by a continuous Laplace distribution. For example, the probability of a quantized parameter from $\boldsymbol{\theta}$ (similarly for $\boldsymbol{\psi}$ and $\hat{W}^i$) is given by:

$$p(\hat{\theta}_k) = \int_{\hat{\theta}_k - 0.5}^{\hat{\theta}_k + 0.5} g(\theta)d\theta, \quad \text{with} \quad g \sim \mathcal{L}(0, \text{stddev}(\hat{\boldsymbol{\theta}})). \tag{5}$$

Then the total rate contribution from neural parameters can be expressed as:

$$R_{\text{MLP}} = \sum_{i=0}^{N} R_{\hat{\boldsymbol{W}}^{\boldsymbol{i}}} + R_{\hat{\boldsymbol{\theta}}} + R_{\hat{\boldsymbol{\psi}}} = -\sum_{k,i} \log_2 p(\hat{W}_k^i) - \sum_k \log_2 p(\hat{\theta}_k) - \sum_k \log_2 p(\hat{\psi}_k). \tag{6}$$

Table 2: Hyper-parameter settings

| Hyper parameter | Initial values | Final values |
|---|---|---|
| **Values of** $\lambda$ | $\{2e^{-2}, 8e^{-3}, 4e^{-3}, 2e^{-3}, 1e^{-3}, 5e^{-4}, 2e^{-4}\}$ | |
| **Quantization – Stage I** | | |
| Number of encoding steps | $10^5$ | |
| Learning rate $\beta$ | $10^{-2}$ | 0 |
| Scheduler for learning rate | Cosine scheduler | |
| Temperature $T$ for soft rounding | 0.3 | 0.1 |
| Noise strength $\alpha$ for Kumaraswamy noise | 2.0 | 1.0 |
| Scheduler for Soft-rounding and Kumaraswamy noise | Linear scheduler | |
| **Quantization – Stage II** | | |
| Number encoding steps | $10^4$ | |
| Learning rate | $10^{-4}$ | $10^{-8}$ |
| Decay learning rate if loss has not improved for this many steps | 40 | |
| Decay factor | 0.8 | |
| Temperature $T$ for soft rounding | $10^{-4}$ | |
| **Architecture – Entropy model** | | |
| Alternative values of $c$ for ARM-$c$ model | $c \in \{8/16/24/32\}$ | |
| Activation function | GELU | |
| Log-scale of Laplace is shifted before $\exp$ | 4 | |
| Scale parameter of Laplace is clipped to | $[10^{-2}, 150]$ | |
| **Architecture – Latent modulations** | **Values** | |
| Number of latent vectors $L$ | 7 | |
| Initialization of $\boldsymbol{z}$ | 0 | |
| **Architecture – Global Modulation Network** | **Single-object vs. Standard** | |
| Upsampling kernel | $4 \times 4$ vs. $8 \times 8$ | |
| Output channels of the $1 \times 1$ convolutions | | |
| $\{7 \to 3/5 \to 3\}$ vs. $\{7 \to 18/24/30 \to 3\}$ | | |
| Output channels of the $3 \times 3$ convolutions | $\{3 \to 3\}$ | |
| Modulation vector dimension | $\{3 \to 3\}$ | |
| **Architecture – LSN** | **Values** | |
| Output dimensions of each layer | $\{2 \to 3 \to 3 \to 3\}$ | |
| Output of each shared modulator (object) | $\{6/8 \to 3\}$ | |
| Out-channels of each shared modulator (standard) | | |
| $\{24/30/36 \to 3\}; \{27/33/39 \to 3\}; \{30/36/42 \to 3\}$ | | |

A greedy search process over quantization steps is conducted to search optimal $\Delta_{W^i}$, $\Delta_\theta$ and $\Delta_\psi$ by minimizing the following RD cost:

$$\min_{\Delta\hat{W}^i,\Delta\hat{\psi},\Delta\hat{\theta}} \sum_{i=0}^{N} \left[ d(\boldsymbol{S^i}, g_{\boldsymbol{W}^i}(f_{\boldsymbol{\theta}}(\hat{\boldsymbol{z}}^{\boldsymbol{i}}), \boldsymbol{x}^i)) + \lambda R(\hat{\boldsymbol{z}}^{\boldsymbol{i}}) \right] + \lambda R_{MLP}. \tag{7}$$

**Model architecture.** We provide the detailed architecture setting for MoRIC in Table. 2.

For the ARM model with $c$ contextual elements as input, denoted as ARM-$c$ model, there are three linear or $1 \times 1$ convolutional layer, followed by GELU activation functions, with input and output dimension given as $(c \times c) \to$ GELU $\to (c \times c) \to$ GELU $\to (c \times 2)$. For the proposed MoRIC, we employ $c \in \{8, 16, 24, 32\}$.

For the GMN model, which uses $L = 7$ layered-resolution latent modulation as the input with $L_m = 4$ layers, the input and output dimension are given as: $(7 \times d) \to$ GELU $\to (d \times 3) \to$ GELU $\to (3 \times 3) \to$ GELU $\to (3 \times 3)$. For the proposed MoRIC, we employ $d \in \{3, 5\}$ for single-object compression and $d = \{18/24/30\}$ for standard image compression such as Kodak and CLIC2020 dataset.

For the LSN model, which maps pixel coordinates to their RGB values with $L_s = 4$ layers, the input and output dimension are given as: $(2 \times 3) \to$ GELU$\to ([3 + m_1] \times 3) \to$ GELU $\to \to (3 \times 3) \to$ GELU $\to \to ([3 + m_2] \times 3) \to$ GELU $\to (3 \times 3) \to$ GELU $\to ([3 + m_3] \times 3) \to$ GELU $\to (3 \times 3)$. Note that here the orange part is shared modulators (one MLP layer). For a PCM design, $m_1 = 3 + d$ (concatenation of first two layer output of GMN), $m_2 = 3 + d + 3$ (concatenation of first three layer output of GMN), $m_2 = 3 + d + 3 + 3$ (concatenation of first four layer output of GMN). For single-object compression, the modulator is shared across layers (then we remove the

---

**Algorithm 2** Encoding stage of the MoRIC

---

**Input:** Source image $\boldsymbol{S}$, coordinate vector for each region $\boldsymbol{x^i}$, Learning rates for Stage I and II: $\alpha$, $\beta$,
Region latent for modulations ($\boldsymbol{z^i}$), and networks (Region LSN $\boldsymbol{W^i}$, GMN $\boldsymbol{\theta^i}$, and ARM $\boldsymbol{\psi}$),
Cosine scheduler for learning rate for Stage I and II: $\mathcal{T}_1$, $\mathcal{T}_2$,
Linear scheduler for soft-rounding temperature $T$ and Kumaraswamy noise $u_{kum}$ strength for Stage I and II: $l_1$,
$l_2$, and Greedy search for quantization step: $\mathcal{G}$.

**Output:** Bits stream of $\boldsymbol{b_z}, \boldsymbol{b_\tau}, \boldsymbol{b_W}, \boldsymbol{b_\theta}, \boldsymbol{b_\psi}$.

1:  $\boldsymbol{S} \rightarrow \{\boldsymbol{S^1}, \ldots, \boldsymbol{S^N}\}; \{\boldsymbol{\tau^1}, \ldots, \boldsymbol{\tau^N}\}; \{\boldsymbol{x^1}, \ldots, \boldsymbol{x^N}\},$   # Partition the source into $N$ regions

2:  $\boldsymbol{z^i} \triangleq \{\boldsymbol{z_1^i}, \boldsymbol{z_2^i}, \ldots, \boldsymbol{z_L^i}\}$ with each $\boldsymbol{z_k^i} \in \boldsymbol{0}^{\frac{K^i}{4^{k-1}}}$   # Initialize the latent for each region

3:  $\boldsymbol{W^i} \sim \mathcal{U}_k, \boldsymbol{\theta} \sim \mathcal{U}_k$   # Configure each LSN and GMN from Kaiming initialization

4:  **for** the $k$-step within the Stage I **do**

5:    **for** each $i$-th region within the $N$ regions **do**

6:     $\hat{\boldsymbol{z}}_i = \mathcal{S}_T(\boldsymbol{z}_i) + \boldsymbol{u}_{kum},$   # Quantization-aware training

7:     $\hat{\boldsymbol{S}}_i = g_{\boldsymbol{W^i}}(f_{\boldsymbol{\theta}}(\hat{\boldsymbol{z}}^i), \boldsymbol{x^i}),$   # Modulate the local reconstruction

8:    **end for**

9:    $\mathcal{L}(\boldsymbol{\Omega}) = \sum_{i=0}^{N}\left[d(\boldsymbol{S^i}, \hat{\boldsymbol{S}}^i) + \lambda R(\hat{\boldsymbol{z}}^i)\right],$   # Compute the RD cost loss function

10:   $\boldsymbol{\theta} \leftarrow \boldsymbol{\theta} - \alpha \nabla_{\boldsymbol{\theta}} \mathcal{L},$   # Update the GMN

11:   $\boldsymbol{z^i} \leftarrow \boldsymbol{z^i} - \alpha \nabla_{\boldsymbol{z^i}} \mathcal{L},$   # Update the latent for each local region

12:   $\boldsymbol{W^i} \leftarrow \boldsymbol{W^i} - \alpha \nabla_{\boldsymbol{W^i}} \mathcal{L},$   # Update the LSN for each region

13:   $\boldsymbol{\psi} \leftarrow \boldsymbol{\psi} - \alpha \nabla_{\boldsymbol{\psi}} \mathcal{L},$   # Update ARM

14:   $\alpha = \mathcal{T}_1(\alpha, k),$   # Schedule learning rate

15:   $T = l_1(T, k), u_{kum} = l_1(u_{kum}, k)$   # Schedule noise strength

16: **end for**

17: **for** the $k$-th step within the Stage II **do**

18:   **for** each $i$-th region within the $N$ regions **do**

19:    $\hat{\boldsymbol{z}}^i = Q(\boldsymbol{z^i}),$   # Hard rounding

20:    $\hat{\boldsymbol{S}}_i = g_{\boldsymbol{W^i}}(f_{\boldsymbol{\theta}}(\hat{\boldsymbol{z}}^i), \boldsymbol{x^i}),$   # Modulate the local reconstruction

21:   **end for**

22:   $\mathcal{L}(\boldsymbol{\Omega}) = \sum_{i=0}^{N}\left[d(\boldsymbol{S^i}, \hat{\boldsymbol{S}}^i) + \lambda R(\hat{\boldsymbol{z}}^i)\right],$   # Compute the RD cost loss function

23:   $\boldsymbol{\theta} \leftarrow \boldsymbol{\theta} - \beta \nabla_{\boldsymbol{\theta}} \mathcal{L},$   # Update the GMN

24:   $\boldsymbol{z^i} \leftarrow \boldsymbol{z^i} - \beta \nabla_{\boldsymbol{z^i}} \mathcal{L},$   # Update the latent for each local region

25:   $\boldsymbol{W^i} \leftarrow \boldsymbol{W^i} - \beta \nabla_{\boldsymbol{W^i}} \mathcal{L},$   # Update the LSN for each region

26:   $\boldsymbol{\psi} \leftarrow \boldsymbol{\psi} - \beta \nabla_{\boldsymbol{\psi}} \mathcal{L},$   # Update ARM

27:   $\beta = \mathcal{T}_2(\beta, k), \beta = \mathcal{T}_2(\beta, k),$   # Schedule learning rate

28:   $T = l_2(T, k), u_{kum} = l_2(u_{kum}, k),$   # Schedule noise strength

29: **end for**

30: $\boldsymbol{b_z} = \sum_{i=0}^{N} \mathcal{A}(Q(\boldsymbol{z^i})),$   # Quantization and entropy coding over $\boldsymbol{z^i}$

31: $\boldsymbol{b_\tau} = \sum_{i}^{N} \mathcal{C}(\boldsymbol{\tau^i}),$   # Adaptive chain coding for contours

32: $\Delta_{\boldsymbol{\theta}}, \Delta_{\boldsymbol{W^i}}, \Delta_{\boldsymbol{\psi}} = \mathcal{G}(\boldsymbol{\theta}, \boldsymbol{W^i}, \boldsymbol{\psi})$   # Search for a optimal quantization step for networks

33: $\boldsymbol{b_\theta} = \mathcal{A}(Q(\boldsymbol{\theta}, \Delta_{\boldsymbol{\theta}}))$   # Quantization and entropy coding over $\boldsymbol{\theta}$

34: $\boldsymbol{b_W} = \sum_{i=0}^{N} \mathcal{A}(Q(\boldsymbol{W^i}, \Delta_{\boldsymbol{W^i}}))$   # Quantization and entropy coding over $\boldsymbol{W^i}$

35: $\boldsymbol{b_\psi} = \mathcal{A}(Q(\boldsymbol{\psi}, \Delta_{\boldsymbol{\psi}}))$   # Quantization and entropy coding over $\boldsymbol{\psi}$

---

---

**Algorithm 3** Decoding stage of the MoRIC

---

**Input:** Bits stream of $\boldsymbol{b_z}, \boldsymbol{b_\tau}, \boldsymbol{b_W}, \boldsymbol{b_\theta}, \boldsymbol{b_\psi}$.

**Output:** Reconstruction of image $\hat{\boldsymbol{S}}$.

1:  $\hat{\boldsymbol{\tau}}^i, \boldsymbol{x^i} \leftarrow \mathcal{C}(\boldsymbol{b_\tau}),$   Chain-decoded contour with its coordinates

2:  $\hat{\boldsymbol{z}}^i = Q(\mathcal{A}(\boldsymbol{b_z})),$   Entropy-decoded and dequantized latent modulation

3:  $\hat{\boldsymbol{\psi}} = Q(\mathcal{A}(\boldsymbol{b_\psi})), \hat{\boldsymbol{\theta}} = Q(\mathcal{A}(\boldsymbol{b_\theta})), \hat{\boldsymbol{W}}^i = Q(\mathcal{A}(\boldsymbol{b_{W^i}})),$   Entropy-deocded network parameters

4:  **for** each $i$-th region within the $N$ regions **do**

5:    $\hat{\boldsymbol{S}}_i = g_{\hat{\boldsymbol{W}}^i}(f_{\hat{\boldsymbol{\theta}}}(\hat{\boldsymbol{z}}^i), \boldsymbol{x^i}),$   # Reconstruct the local region

6:  **end for**

7:  $\hat{\boldsymbol{S}} = \bigcup_{i=1}^{N} \hat{\boldsymbol{S}}^i,$   Merge the source image

---

concatenation for $m_1 = m_2 = m_3 = d$ for less parameters); for standard image compression, the modulator is shared across regions.

**Pseudocode for the algorithm.** This section provides the encoding and decoding procedures of MoRIC, presented in Algorithm 2 and Algorithm 3, respectively. We use orange to indicate GMN-related optimization, purple for ARM-related optimization, blue for local region-specific optimization, red for chain coding, and green for LSN optimization.

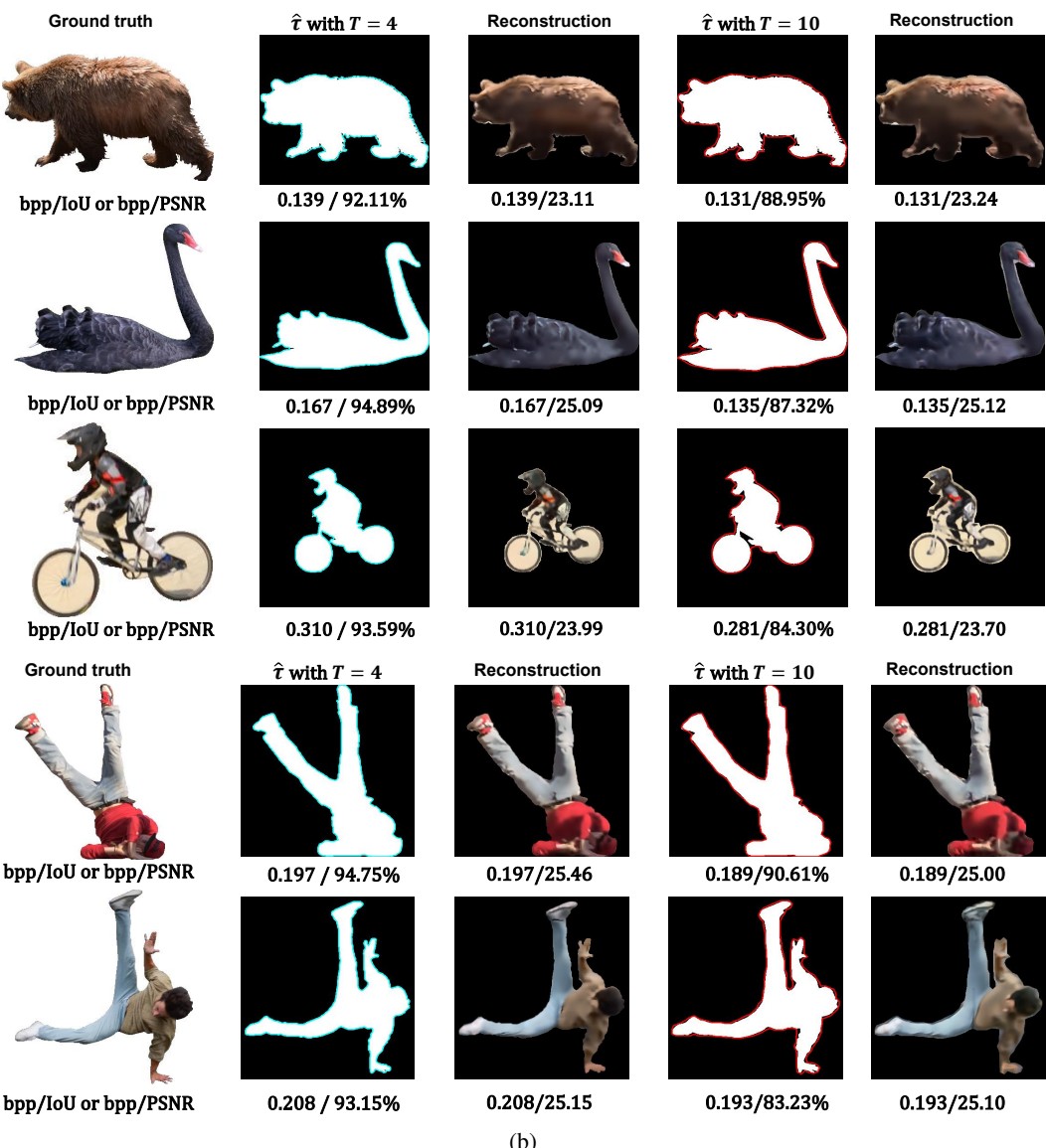

(b)

Figure 13: Visualization of the proposed $C^*$ coding under varying step lengths $T$ (larger $T$ resulting in coarser contours) in the **low-bpp regime** ($\lambda = 4e^{-2}$), with corresponding bpp, IoU, and PSNR.

# E    Additional experiments and visualizations

## E.1    Contour compression using $C^*$ coding

This section presents visualizations of the proposed $C^*$ coding under varying step lengths $T$ across different bitrate. C* coding approximates object contours using polygonal chains. Starting from an initial point, each contour segment is encoded by a fixed step length and a quantized offset angle relative to the previous direction.

As shown in Fig. 13, from the view of RD performance, in the low-bpp regime, coarse $C^*$ coding achieves comparable PSNR with significantly lower bit cost. Similarly, a high-bpp regime result is visualized in Fig. 14. We can see that for high-bpp regime, a more-accurate contour compression approach is affordable, resulting in a better RD performance. We want to note that, the adaptive step length $T$ allows flexibility to meet user-specific needs. For instance, users prioritizing semantic accuracy or precise contours can opt for a smaller $T$ at the expense of additional bits. Additional experiments about the chain-coding are also provided in Table 3.

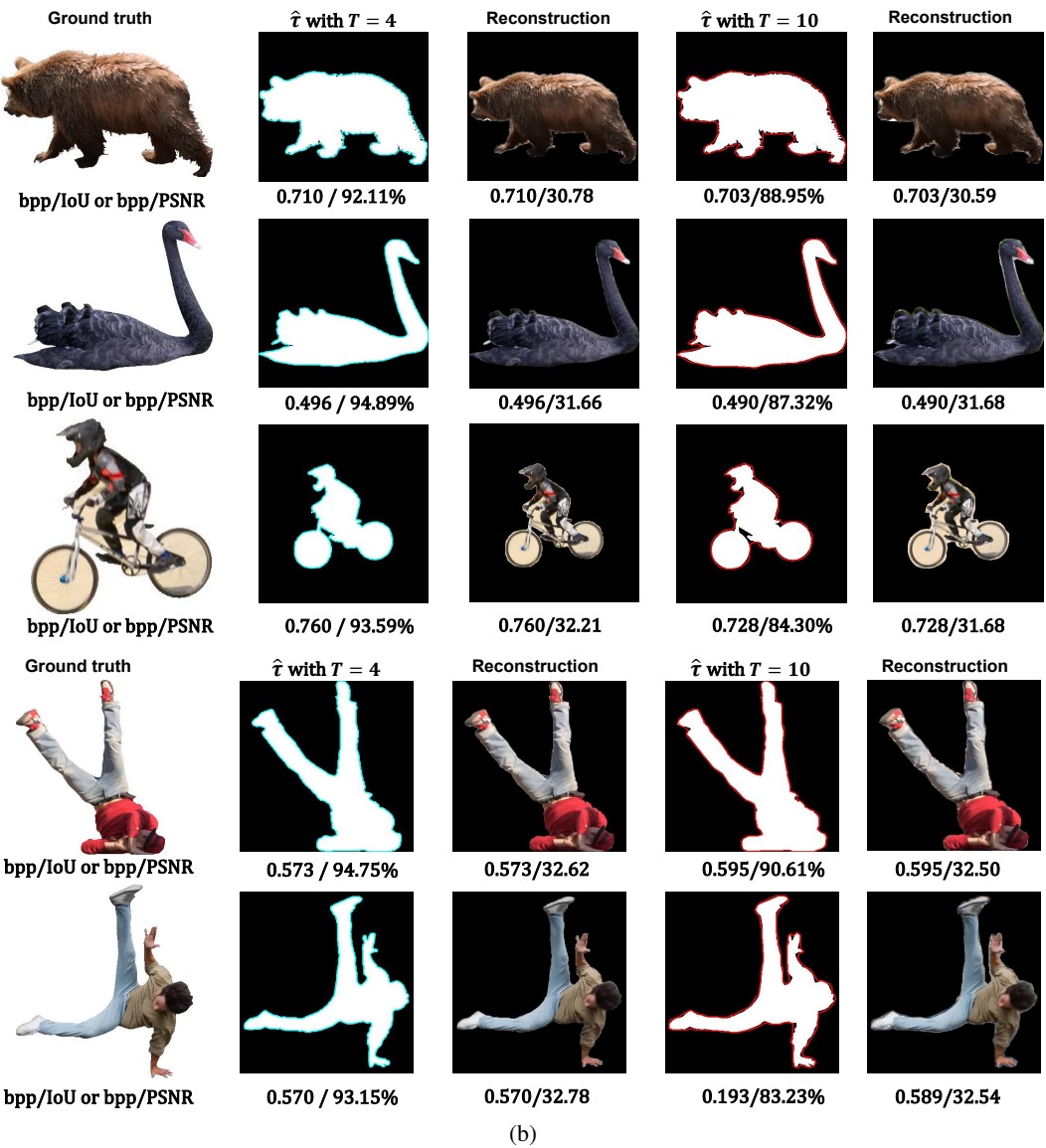

Figure 14: Visualization of the proposed $C^*$ coding under varying step lengths $T$ in the **high-bpp regime** ($\lambda = 2e^{-3}$), with corresponding bpp, IoU, and PSNR.

Table 3: Contour coding efficiency on Kodak dataset (kodim23), evaluated by error rate/bit cost.

| Lossless chain coding | 0%/1641 | | |
|---|---|---|---|
| **Quality values of VTM** | 10 | 36 | 56 |
| **Performance of VTM** | 0%/9128 | 1.48%/6864 | 6.62%/1960 |
| **Step length of C\*** | 5 | 10 | 15 |
| **Performance of C\*** | 0.66%/890 | 1.17%/360 | 2.08%/251 |

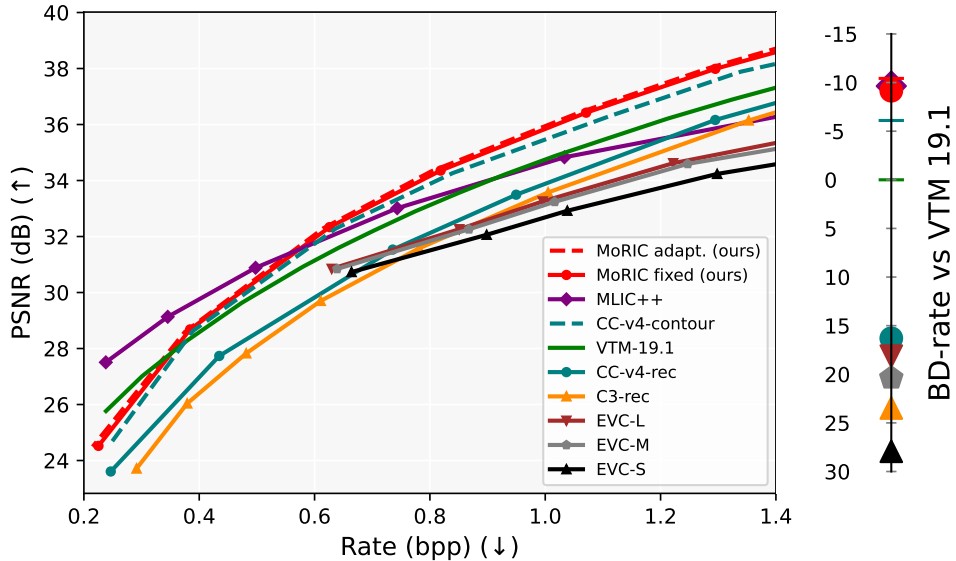

Figure 15: RD curve and BD-rate with more baselines for **single-object** compression on Davis.

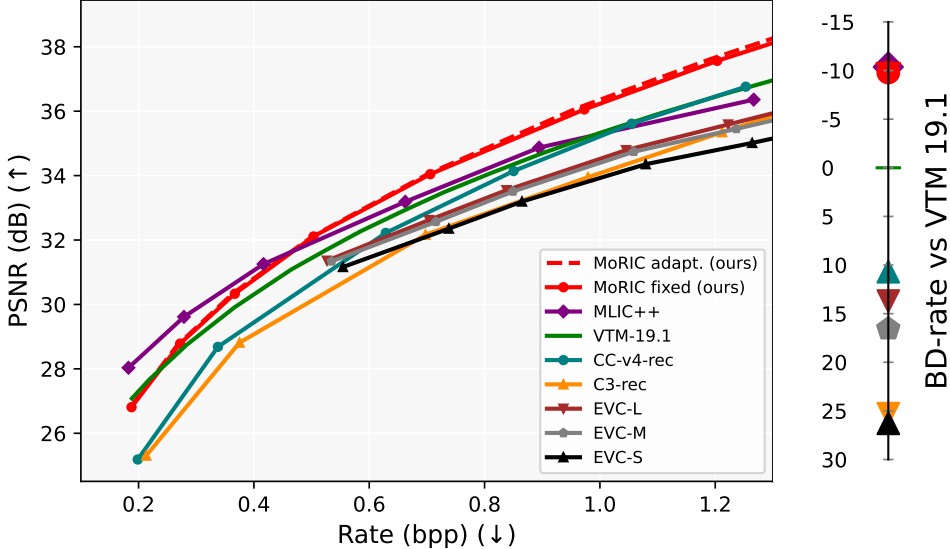

Figure 16: RD curve and BD-rate with more baselines for **single-object** compression on Kodak.

## E.2 Single-object compression

**Experiment setup**    Single-object compression task only considers the compression over a target region. Specifically, for AE-based codecs in the single-object compression task, the target object is cropped out using the smallest enclosing rectangle, with all pixels outside the target region set to zero for baseline methods, effectively allowing AE-based codecs to implicitly encode the contour and achieve optimal performance. For overfitted codecs, we evaluate (1) COOL-CHIC-v4 combined with our precise contour coding (denoted as CC-v4-contour) to assess architectural gains, and (2) C3/CCv4 combined with a minimal bounding-box contour (C3-rec; CCv4-rec) as an alternative baseline.

**Additional experiments**    For more detailed experiments for single-object compression task, Figs. 15-16 and Table 4 present detailed RD performance, along with ablations on applying our chain coding to other overfitted codecs and adaptive configurations, which further validates the effectiveness of our design. Additional results of decoding complexity, encoding latency, and BD-rate are provided

Table 4: Detailed BD rate vs. VTM-19.1 for **single-object** compression task.

| Dataset | MoRIC | MLIC$^{++}$ | EVC-S | EVC-M | EVC-L | CC V4 | C3 |
|---|---|---|---|---|---|---|---|
| Davis dataset | $-\mathbf{10.43}\%$ | $-9.65\%$ | $27.92\%$ | $20.39\%$ | $18.14\%$ | $16.31\%$ | $23.48\%$ |
| Kodak dataset | $-\mathbf{10.73}\%$ | $-10.33\%$ | $26.28\%$ | $16.57\%$ | $13.70\%$ | $10.69\%$ | $25.29\%$ |

in Fig.17 and Table 5. We can see that MoRIC can achieve faster decoding but slightly slower encoding in terms of latency for single-object compression.

Table 5: Average coding time for **single-object compression** on the DAVIS dataset using an NVIDIA RTX 3090 GPU and Intel Core i9-10980XE CPU @ 3.00GHz. Orange indicates GPU computation, blue indicates CPU computation, and bold numbers highlight the best results.

| Models | Encoding time | Decoding time | BD rate |
|---|---|---|---|
| VTM-19.1 | 21.53 s | 136.02 ms | 0 |
| EVC (S/M/L) | **10.62/13.72/27.71** ms | 9.41/16.18/18.48 ms | $27.92\%/20.39\%/18.14\%$ |
| MLIC$^{++}$ | 89.71 ms | 145.20 ms | $-9.65\%$ |
| C3 | 11.87 sec / 1k steps | 141.48 ms | $+23.47\%$ |
| COOL-CHIC | 14.28 sec / 1k steps | 136.83 ms | $+16.31\%$ |
| MoRIC | 17.54 sec / 1k steps | **102.53** ms | $-\mathbf{10.43}\%$ |

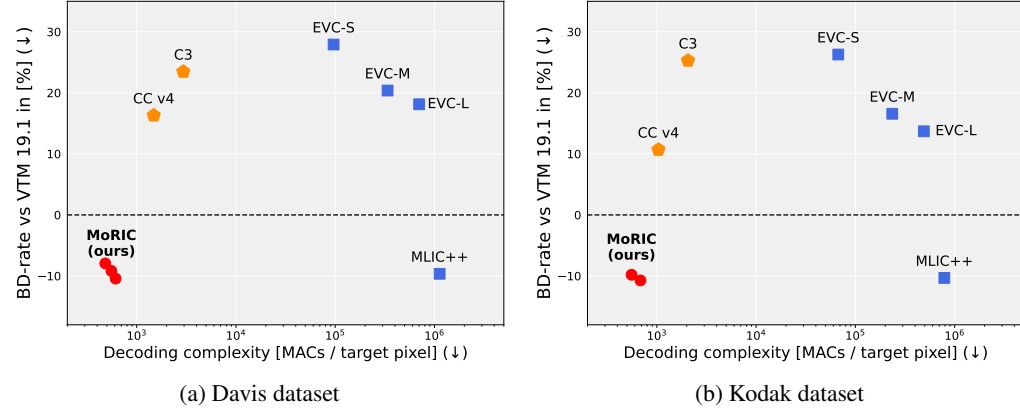

(a) Davis dataset         (b) Kodak dataset

Figure 17: BD-rate vs. decoding complexity for **single-object** compression.

In summary, without incurring high costs and implicit contour learning, MoRIC enables scalable and practical single-object compression with significantly improved RD performance and lowering decoding complexity. This is especially promising for fixed-background scenarios such as surveillance, where objects can be precisely compressed while background is reused across frames to reduce the overall bitrate.

**Visualization** We additionally provide visualizations of MoRIC for the single-object compression task, as shown in Figs. 18 and 19. Compared with MLIC++, in addition to better RD performance, MoRIC retains more image details (see the dog's leg in the picture as an example).

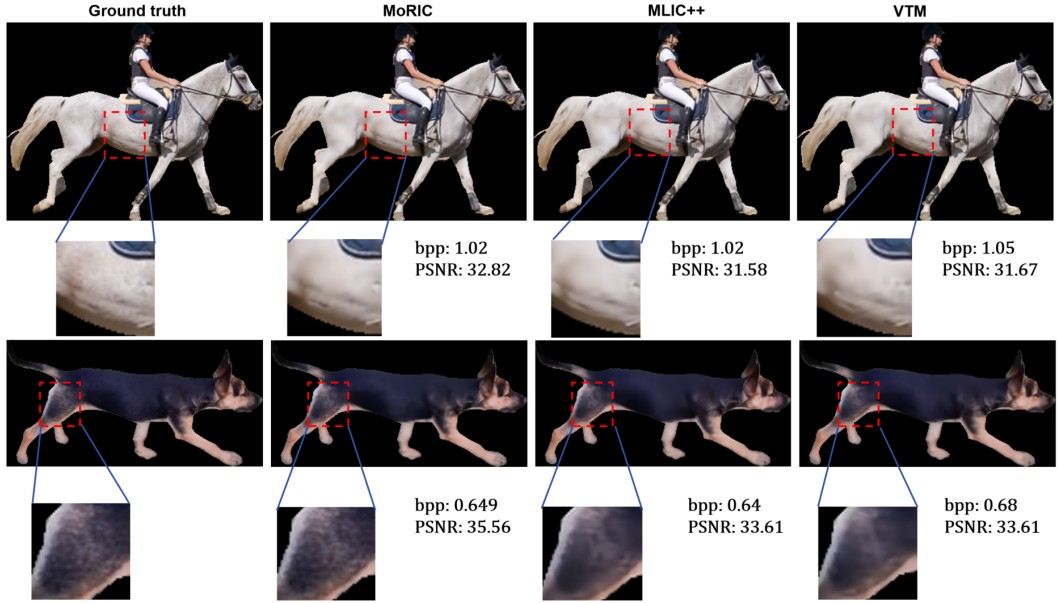

Figure 18: Visualization of **single-object** compression on Davis.

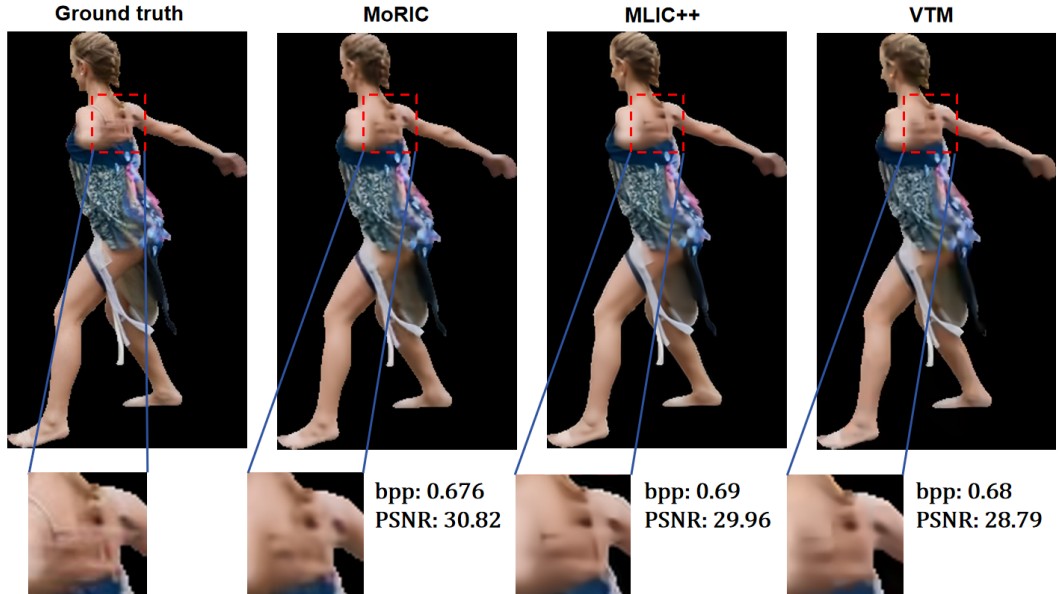

Figure 19: Visualization of **single-object** compression on Davis.

### E.3  Standard full image compression

For more experiments on the standard full image compression task, more detailed RD-curves are given in Figs. 20, 21, 22, and Table 6, 7, with raw datapoints presented in Table 8. Detailed experimental results over the effect of region number is given in Fig. 24, with visualization provided in Fig. 25 and Table 9. The effect of latent vectors are visualized in Figs. 26, 27.

Complexity and latency comparisons are provided in Table. 10 and Fig. 9. Note that all the results are based on unoptimized research code and the available hardware; performance is expected to improve with engineering optimizations such as C-API and wavefront decoding. The BD-rate of MoRIC (three-region setting) in Table 10 is evaluated using a mix of $1, 2, 3$ region settings, as only 10 images

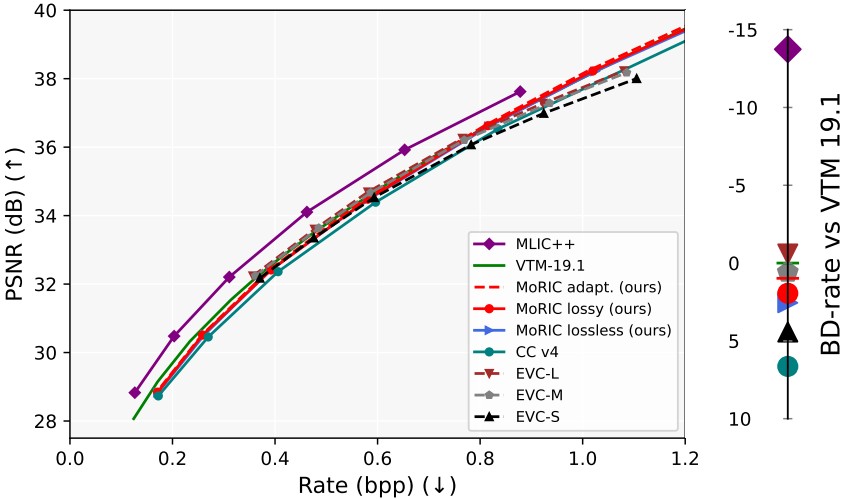

Figure 20: RD and BD-rate with more baselines for **standard full image compression** on Davis.

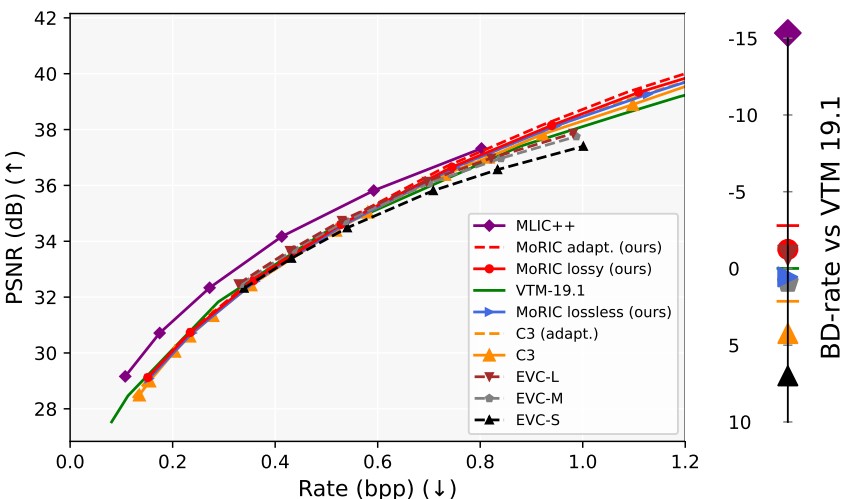

Figure 21: RD and BD-rate with more baselines for **standard full image compression** on Kodak.

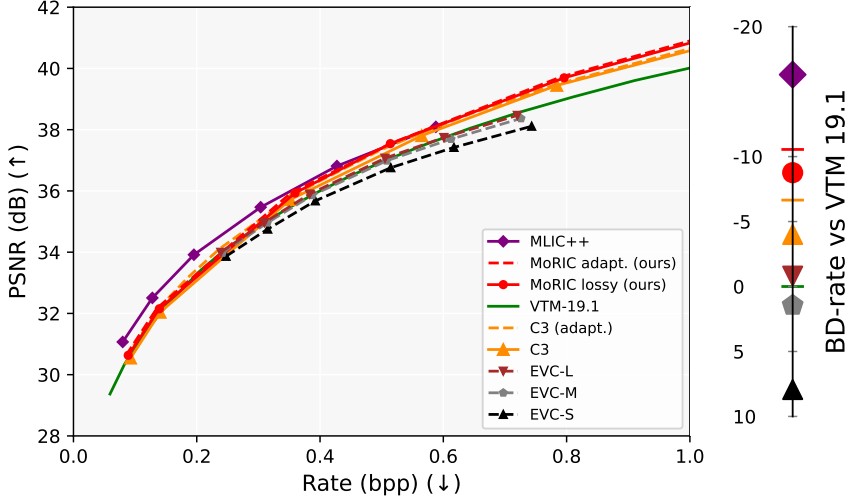

Figure 22: RD and BD-rate with more baselines for **standard full image compression** on CLIC2020.

in the dataset were annotated with three-region masks. Further improvements can be expected with full-dataset annotations.

Overall, our method has a slightly longer encoding time than other overfitted codecs due to its more advanced modulation mechanism, but delivers superior performance. We want to note that all reported MoRIC results are based on unoptimized research code. Notably, our decoding modules are now executed serially during latency evaluation; a simple parallel execution can substantially reduce runtime. We also expect further improvements through advanced engineering optimizations such as C-API integration and binary arithmetic coding

Table 6: Detailed BD rate vs. VTM-19.1 for **standard full image compression** task.

| Dataset | MoRIC (lossy-adapt.) | MoRIC (lossy-fixed) | MoRIC (lossless) | MLIC$^{++}$ | EVC SS/MM/LL | COOL-CHIC v4 |
|---|---|---|---|---|---|---|
| Davis | **0.981**% | **1.94**% | 2.55% | $-13.73$% | 4.43% / 0.59% / $-0.58$% | 6.64% |
| Dataset | MoRIC (lossy-adapt.) | MoRIC (lossy-fixed) | MoRIC (lossless) | MLIC$^{++}$ | EVC SS/MM/LL | C3 adapt. |
| Kodak | **$-2.79$**% | **$-1.24$**% | 0.78% | $-14.96$% | 7.16% / 1.05% / $-0.73$% | 2.64% |

Table 7: More baselines on Kodak and CLIC datasets.

| Kodak dataset | | | | | |
|---|---|---|---|---|---|
| **Traditional/INR codec** | **JEPG2000** | **HEVC (HM)** | **BPG** | **MSCN** | **VC-INR** | **RECOMBINER** |
| BD-rate over VTM-19.1 | 95.42% | 24.90% | 22.53% | 115.91% | 73.71% | 63.86% |
| **Autoencoder codec** | **BMS** | **MBT** | **CST** | **WYH** | **STF** | **ELIC** |
| BD-rate over VTM-19.1 | 30.34% | 10.85% | 3.99% | $-0.44$% | $-3.96$% | $-6.85$% |
| **CLIC2020 dataset dataset** | | | | | |
| **Image codec** | **BPG** | **ELIC** | **Cheng2020** | **WYH** | **STF** | **CCv4** |
| BD-rate over VTM-19.1 | 39.28% | $-7.04$% | 7.22% | $-4.77$% | $-3.22$% | $-6.58$% |

Table 8: Raw datapoints of MoRIC for standard full image compression task.

| Rate [bpp] | PSNR [dB] | Rate [bpp] | PSNR [dB] | Rate [bpp] | PSNR [dB] |
|---|---|---|---|---|---|
| 0.1507 | 29.118 | 0.0852 | 30.628 | 0.169 | 28.834 |
| 0.3533 | 32.622 | 0.1354 | 32.168 | 0.257 | 30.503 |
| 0.5217 | 34.637 | 0.3534 | 35.942 | 0.390 | 32.428 |
| 0.7339 | 36.701 | 0.5141 | 37.544 | 0.577 | 34.503 |
| 1.0851 | 39.353 | 0.7898 | 39.715 | 0.809 | 36.629 |
| 1.4173 | 41.307 | 1.0722 | 41.306 | 1.010 | 38.228 |
| | | | | 1.196 | 39.511 |
| | | | | 1.559 | 41.793 |
| (a) Kodak dataset | | (b) CLIC2020 dataset | | (c) Davis dataset | |

Table 9: Bit rate breakdown for MoRIC across region numbers on Kodak dataset.

| Bit cost share | Contour (%) / Parameters (%) / Latent (%) | | | BD-rate vs. MoRIC (single) |
|---|---|---|---|---|
| | Low bpp | Medium bpp | High bpp | |
| **MoRIC (single region)** | 0.11815 0%/25.39%/74.61% | 0.41885 0%/8.14%/91.86% | 1.27234 0 / 2.89% / 97.11% | 0 |
| **MoRIC ($N = 2$ regions)** | 0.11940 1.89%/26.01%/72.09% | 0.41879 0.54%/8.41%/91.05% | 1.24859 0.18%/3.11%/96.71% | -1.29% |
| **MoRIC ($N = 3$ regions)** | 0.12067 3.20%/26.71%/70.08% | 0.41567 0.93%/8.55%/90.52% | 1.24135 0.31%/3.22%/96.47% | -1.72% |

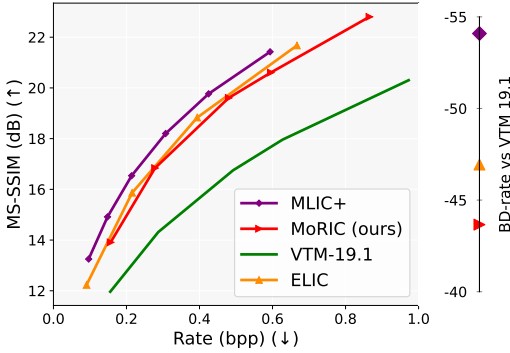

Figure 23: RD curve with MS-SSIM distortion and BD-rate for **standard full image compression** on the Kodak.

Figure 24: RD curve and BD-rate with various number of regions for **standard full image compression** on the Kodak, where the test set is 10 images and a fixed architecture is employed.

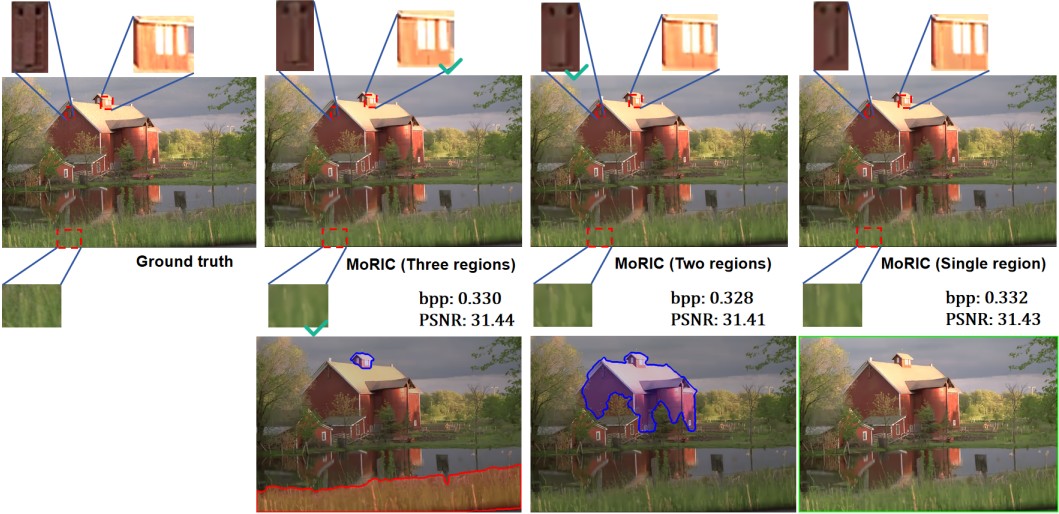

Figure 25: Effect visualization of the region number for MoRIC: The top row shows reconstruction results, while the bottom row displays the corresponding region masks. As the number of regions increases (adding a dedicated region over the roof and grass as an example), more detailed reconstructions are achieved in those specific areas. In contrast, the single-region setting produces relatively coarser reconstructions compared to multi-region configurations.

Table 10: Average coding time for **standard full image compression** on Kodak dataset using an NVIDIA RTX 3090 GPU and Intel Core i9-10980XE CPU @ 3.00GHz. Orange indicates GPU computation, blue indicates CPU computation, and bold numbers highlight the best results.

| Models | Encoding time | Decoding time | BD rate |
|---|---|---|---|
| VTM-19.1 | 87.13 s | 293.05 ms | 0 |
| EVC (S/M/L) | **22.43/36.19/47.87** ms | **19.89/24.12/31.91** ms | $7.16\%/1.05\%/-0.73\%$ |
| MLIC$^{++}$ | 271.91 ms | 364.06 ms | $\mathbf{-14.96\%}$ |
| C3 (fixed / adapt) | 17.49/21.33 sec / 1k steps | **283.59 / 310.54** ms | $+4.75\%/+2.64\%$ |
| MoRIC (single region) | 24.70 sec / 1k steps | 343.55 ms | $0.54\%$ |
| MoRIC (two region) | 26.41 sec / 1k steps | 408.97 ms | $-2.79\%$ |
| MoRIC (three region) | 32.06 sec / 1k steps | 486.48 ms | $-2.88\%$ |

### E.4 Layered compression

**Layered coding strategy**   Towards a layered compression scheme, MoRIC first transmits background INRs (LSN-0 for layer-0 transmission) to reconstruct coarse structures with some detail

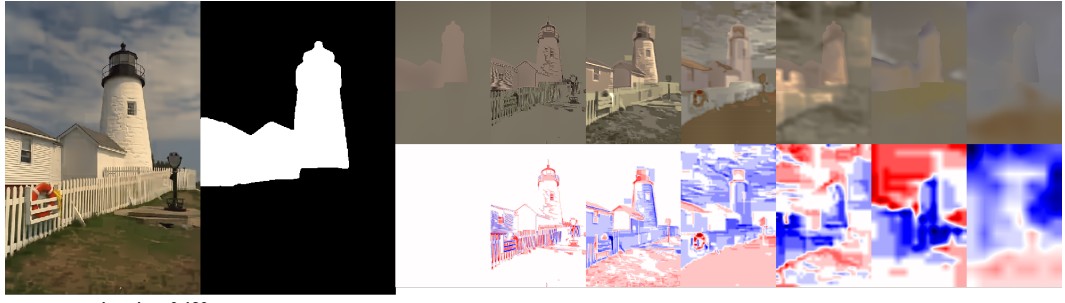

Low bpp 0.123

Figure 26: Reconstruction and visualization of latents for kodim-19 at low (0.123 bpp) bitrates. The first row shows reconstructions with only one of seven latent modulations sets active at a time. The second row visualizes the corresponding raw latents, upsampled to output resolution.

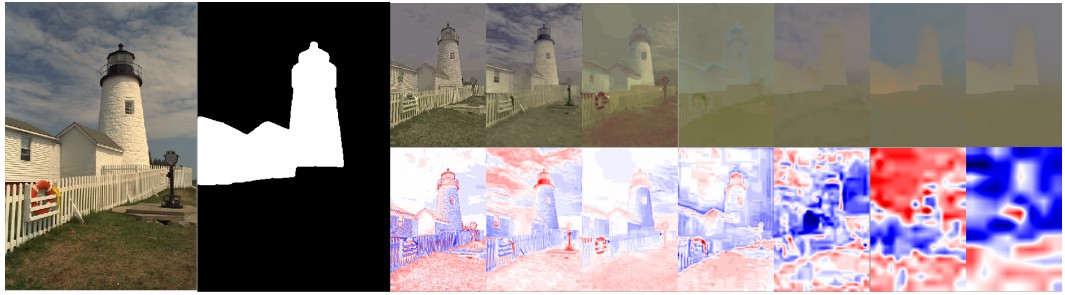

High bpp 1.425

Figure 27: Reconstruction and visualization of latents for kodim-19 at high (1.425 bpp) bitrates. The first row shows reconstructions with only one of seven latent modulations sets active at a time. The second row visualizes the corresponding raw latents, upsampled to output resolution.

inaccuracies (e.g., object colors) while preserving semantic correctness. This aligns with our $C^*$ coding design, which prioritizes accurate object boundaries while allowing coarser approximations for background regions to preserve semantic integrity and reduce artifacts. Subsequently, to refine specific regions, corresponding LSNs (LSN-1) are transmitted as layer-1. The remaining areas are inferred jointly from both layers (LSN-0 and LSN-1) through averaging. This process continues with additional LSNs, enabling progressive refinement and extensibility to more layers/LSNs.

**More experimental examples** To further illustrate the semantic generalization capability for layered compression application, we use the model trained on the Kodak dataset with three regions (Fig. 24) to perform inference across different regions using different LSNs.

Results are visualized in Fig. 28, 29, 30, we can observe that local INRs can infer semantics beyond their assigned regions. By progressively transmitted LSN for different regions, it can refine earlier outputs. This behavior comes from the global-local modulation design: LSNs can capture shared features across regions (e.g., common colors) while encoding contrastive, region-specific details (e.g., distinct object colors) that are unrecoverable by other LSNs.

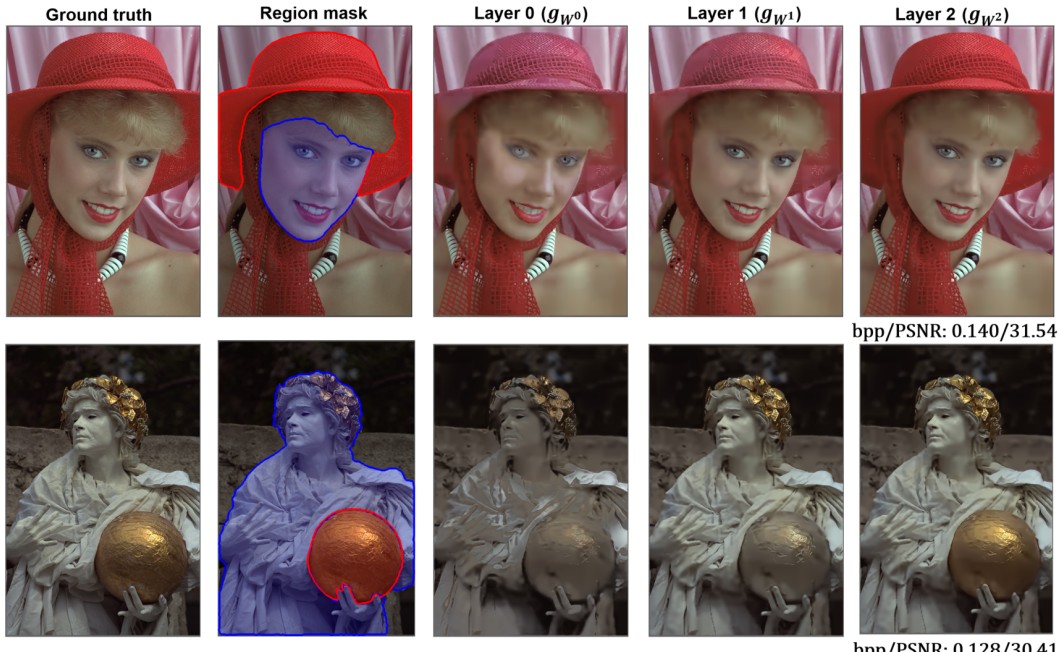

Figure 28: Visualization of layered compression: The background LSN $g_{\boldsymbol{W}^0}$ is first transmitted as layer 0, followed by $g_{\boldsymbol{W}^1}$ for the blue region (layer 1), and finally $g_{\boldsymbol{W}^2}$ for the red region (layer 2). We can see that colors and textures (e.g., red hat and golden ball) are refined with the additional LSN parameters.

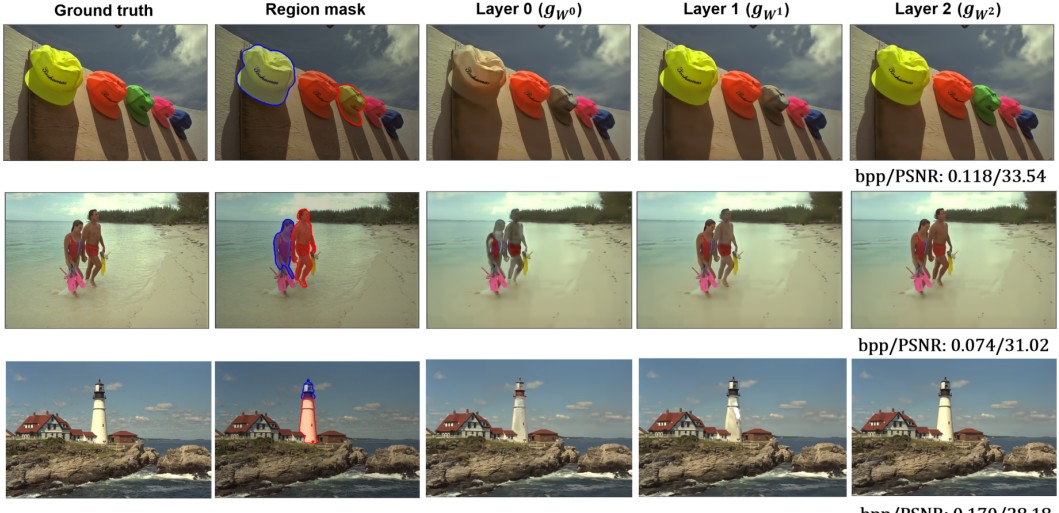

Figure 29: Visualization of layered compression: The background LSN $g_{\boldsymbol{W}^0}$ is first transmitted as layer 0, followed by $g_{\boldsymbol{W}^1}$ for the blue region (layer 1), and finally $g_{\boldsymbol{W}^2}$ for the red region (layer 2). We can see that specific colors (yellow and green hat / people / tower) and textures are refined with the additional LSN parameters.

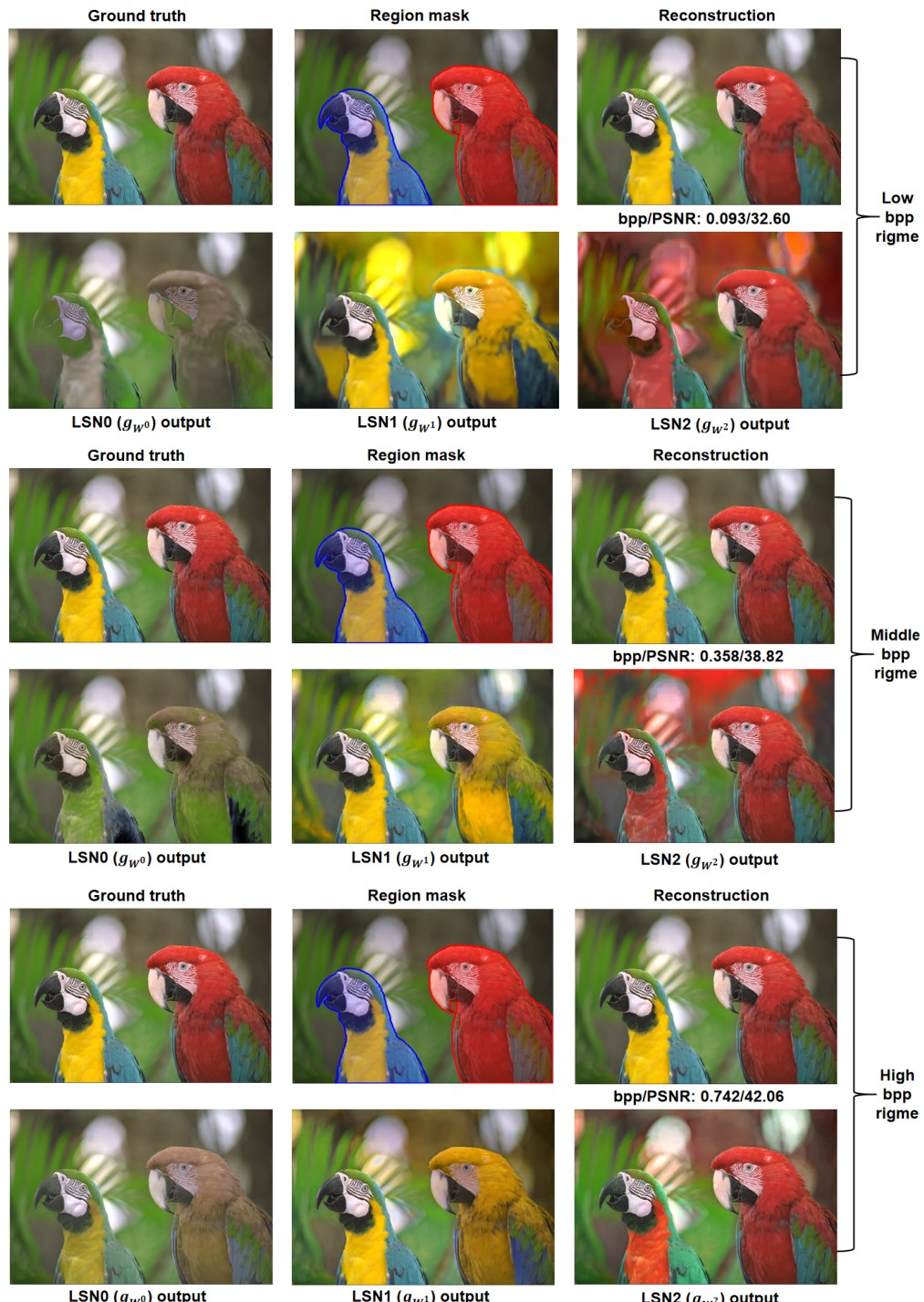

Figure 30: Visualization of layered compression at various bitrates, where the background LSN is transmitted as layer 0, followed by the blue and red region LSNs as layers 1 and 2. Shared features (e.g., green color, bird mouth) are captured across regions, while contrastive details (e.g., bird-specific colors) are refined by region-specific LSNs. At low bpp, inferring context and features from other regions is more challenging; at high bpp, richer latent modulations enable better cross-region inference.

## E.5   Experiments for partial decoding

To further illustrate the benefits of partial decoding, we report detailed computation and bit cost savings for a surveillance scenario in Table 11. For experimental details, we refer to Fig. 5 in our supplementary materials for a demo illustration. We can observe that MoRIC achieves up to 73.56% and 37.31% bitrate savings compared to VTM and MLIC++, respectively. Thanks to the use of contour information, MoRIC achieves significantly lower decoding complexity in partial decoding scenarios by coding only the target region with simpler neural functions. As shown in Table 11, baseline methods require approximately 2x (VTM) and 1235x (MLIC++) more computation than MoRIC.

Table 11: Bitrate and computational savings enabled by partial decoding in MoRIC.

| Method | VTM | MLIC++ | MoRIC |
|---|---|---|---|
| PSNR(dB) | 52.301 | 52.332 | 52.584 |
| bit cost ($10^4$) | 2.107 | 1.667 | 1.214 |
| bit saving vs. methods | 73.56% | 37.31% | - |
| Computation saving vs. methods | 2 times | 1235 times | - |

## E.6   Ablation study over model components

To evaluate each component of MoRIC: $C^*$ coding, GMN with PCM, and LSNs, this section presents ablation studies on each component. The results are reported in Table 12, each component contributes to the performance of MoRIC. $C^*$ coding is essential for enabling region-wise compression, yielding a $-0.47\%$ BD-rate reduction. The PCM module contributes a significant $-3.56\%$ decrease in BD rate, highlighting its impact on RD performance. Removing the LSN reduces MoRIC to a variant of the C3/COOL-CHIC series, resulting in RD performance decrease ($+5.35\%$ BD-rate), underscoring the importance of modular region-wise design.

| Model Variant | BD rate vs. MoRIC |
|---|---|
| *Proposed* MoRIC scheme | 0.0% |
| $\Rightarrow$ w/o $C^*$ coding (rectangular region) | +0.47% |
| $\Rightarrow$ w/o PCM-GMN | +3.56% |
| $\Rightarrow$ w/o LSN | +5.35% |

Table 12: Ablation study on different mechanisms for **standard full image compression**, using an ARM-16 model, where $\lambda \in \{8e^{-3}, 2e^{-3}, 1e^{-3}, 5e^{-4}, 3e^{-4}, 2e^{-4}\}$. A higher BD rate indicates worse RD performance.

## E.7   Ablation study over artifacts

For a region-based coding scheme, the contour boundaries of MoRIC remain barely visible, even at low bitrates such as 0.05 bpp (see our visualizations). This robustness stems from two key design choices in MoRIC:

- Design of the global modulation network (GMN): MoRIC integrates global-local context through its GMN and progressive modulation, effectively suppressing artifacts. Similar strategies have been shown to reduce artifacts in previous INR literatures [1, 19].
- Region-wise fitting strategy: Instead of fitting the image in a patch-wise manner, MoRIC performs region-wise fitting, enabled from C* coding, enhancing INR representation while reducing artifacts (as also observed in [1]).

To further quantify the effect of artifacts, we include an ablation study on GMN and region-wise fitting (exp-1 and exp-2) for artifacts analysis and quality assessments, reporting PSNR, MS-SSIM,

PSNR-edge/PSNR-smooth edge in Tables 1314 (two metrics from [19] that assess artifacts via Canny edge detection with disk dilation). For exp-1, we compare MoRIC with a variant that removes the GMN component, denoted as MoRIC (w/o GMN). Exp-2 replaces the target region of MoRIC with a rectangular patch to cover the same target, denoted as MoRIC (rect), where contour coding is omitted.

Table 13: Ablation study for MoRIC (w/o GMN) for artifacts analysis on Kodak.

| Rate | 0.1543 | 0.5399 | 0.7892 | 1.2003 | BD-rate vs. MoRIC (%) ↓ | Observed artifacts |
|---|---|---|---|---|---|---|
| PSNR | 28.910 | 34.437 | 36.445 | 39.057 | +7.03 | Without GMN: |
| MS-SSIM | 11.403 | 16.788 | 18.674 | 21.108 | +6.91 | Ringing artifacts, |
| $PSNR_{edge}$ | 25.038 | 32.104 | 34.527 | 37.604 | +6.11 | gradient discontinuity, |
| $PSNR_{smooth}$ | 31.960 | 35.628 | 37.326 | 39.635 | +8.56 | inconsistent textures |

Table 14: Ablation study for MoRIC (rect) for artifacts analysis on Kodak.

| Rate | 0.1500 | 0.5399 | 0.7496 | 1.1157 | BD-rate vs. MoRIC (%) ↓ | Observed artifacts |
|---|---|---|---|---|---|---|
| PSNR | 29.067 | 34.509 | 36.645 | 39.317 | +2.53 | With rec contours: |
| MS-SSIM | 11.518 | 16.911 | 18.857 | 21.273 | +2.28 | Very visible |
| $PSNR_{edge}$ | 25.159 | 32.122 | 34.747 | 37.950 | +2.87 | stitching artifacts |
| $PSNR_{smooth}$ | 32.171 | 35.720 | 37.490 | 39.817 | +2.02 | between patches. |

## E.8 Ablation study over segmentation errors

We conducted additional experiments by training MoRIC using masks with varying degrees of over- and under-segmentation errors. Specifically, we simulate small and substantial segmentation errors via mask dilation and erosion, with IoU values of $14.22\%/34.18\%$ (under-segmentation) and $19.85\%/48.58\%$ (over-segmentation).

Results are shown in the table below. We observe that MoRIC is robust to segmentation imperfections, especially when the segmentation IoU is below $20\%$, with performance degrading gradually as segmentation quality decreases. However, for both under- and over-segmentation, more significant performance drops begin to appear when the IoU approaches $30\check{}40\%$. Additionally, substantial segmentation errors can introduce encoding instability, occasionally leading to suboptimal RD performance. While MoRIC does not currently optimize region selection, we believe that better segmentation strategies could yield substantial improvements by leveraging this trade-off.

Table 15: **Ablation study over segmentation error for MoRIC on Kodak dataset.**

| Methods | | PSNR (dB) / bpp | | | | BD-rate |
|---|---|---|---|---|---|---|
| MoRIC (lossy C* coding) | 3.95% IoU | 29.121/0.1516 | 34.599/0.5279 | 36.655/0.7449 | 41.259/1.4533 | 0 |
| Under-segmentation | 14.22% IoU | 29.035/0.1537 | 34.573/0.5293 | 36.570/0.7664 | 41.256/1.4609 | +2.11 |
| | 34.18% IoU | 29.059/0.1530 | 34.569/0.5323 | 36.602/0.7604 | 41.042/1.5078 | +2.76 |
| Over-segmentation | 19.85% IoU | 29.066/0.1541 | 34.564/0.5327 | 36.624/0.7617 | 41.235/1.4666 | +2.09 |
| | 48.58% IoU | 29.038/0.1553 | 34.562/0.5327 | 36.603/0.7591 | 41.064/1.5513 | +3.29 |

