# OpenReview forum: "MoRIC: A Modular Region-based Implicit Codec for Image Compression"
_NeurIPS.cc/2025/Conference — NeurIPS 2025 poster_

### Official Review · Reviewer_1oDR · 2025-06-21

**Clarity:** 3
**Significance:** 3
**Originality:** 3
**Rating:** 5
**Confidence:** 3

**Summary:**

The paper proposes MoRIC, an implicit image compression method consisting of multiple (N=3 is the default) local compressors, each optimized for a specific image sub region specified by a contour. The method is evaluated on 2 datasets for image compression single-object compression. The method achieves SotA in single-object compression for bpp>0.5 and good performance in image compression.

**Questions:**

- The lossy contour compression approximates the contour with a polygon, did I understand this correctly? If so, I would add a sentence somewhere explicitly stating this, the whole section reads a bit too abstract.
- Are contour boundaries very visible? I looked at the full image compression examples in the appendix but the images are too small. I would recommend adding an example for very low bitrate that shows what kind of boundary artifacts can be expected
-  Since in an actual application the GT masks might not be available it would be interesting what the effect of incorrect segmentation masks is on the overall performance. Did you look into this?

**Ethical Concerns:**

["NO or VERY MINOR ethics concerns only"]

**Final Justification:**

The paper presents an interesting approach to region based compression. It is well written and evaluated and performs well. The additional evaluations on CLIC, a common compression dataset together with the analysis of the impact of minor segmentation errors made me raise my score to "accept".

**Limitations:**

yes

**Paper Formatting Concerns:**

no formatting concerns

**Quality:**

3

**Strengths And Weaknesses:**

**Strengths:**
- The paper is well written and contains multiple well designed figures
- The method contains multiple novel ideas, such as contour compression, using different networks for different regions and the global modulation network, a network with shared weights among different regions that processes the latent and helps incorporate global information
- The experiments are very extensive, comparing to multiple SotA methods and an ablation study

**Weaknesses:**
- For the contour GT information was used (for Davis) and for Kodak the contours were "manually selected"(?). In a real compression setting GT masking information can not be expected making this a purely academic method.
- The model is still outperformed by MLIC++ in image compression and even for single-object compression for low bpp (<0.5)
- While the decoding complexity is lower than other learned codecs the decoding time (which is practically often the more important metric) is higher (see tab. 6)
- Davis is somewhat of a non-standard choice for benchmarking, which might make it more difficult to compare the method to other methods in the future. CLIC is mentioned in the supplements but the results are neither in the main paper nor in the supplements?

---

> ### Author Rebuttal · Authors · 2025-07-31
>
> We thank the reviewer for their valuable feedback. Here is our point-to-point response:
>
> **Response to each weakness:**
>
> - **W1.** We currently use SAM2 for region partitioning, which **inherently supports automatic segmentation** and can be extended with other advanced strategies such as saliency-based or prompt-guided segmentation for various applications. For alternative partition methods, we refer the reviewer to our **response to [3Cij]** (W1. and Q3.)
>
> - **W2.** The reviewer is correct. We also want to note that while MoRIC falls short of the MLIC++ in RD performance for standard image compression, it achieves over 100× lower decoding complexity. The superior performance of MLIC++ in the low-bpp regime for single-object scenarios can be attributed to its natural advantage in this setting. In contrast, overfitted codecs face inherent limitations due to the dominance of parameter cost in this regime.
>
>   In response to this comment, we have added the following clarification in the revised manuscript: "Although MoRIC shows significant RD gains, it still shares the common low-bpp performance bottleneck with other overfitted codecs due to dominant parameter cost, a challenge that Bayesian INR parameter [1] coding may help address."
>
>    [1] He, Jiajun, et al. "RECOMBINER: Robust and Enhanced Compression with Bayesian Implicit Neural Representations."
>
> - **W3.** We thank the reviewers for pointing out this issue. While MoRIC has lower theoretical decoding complexity, its observed latency remains slightly high due to the lack of engineering-level optimizations. We want to note that real-world latency is affected by many uncontrollable factors in a lab setting and can be significantly reduced through various optimization techniques. For example, [2] reports 100 ms decoding time for COOL-CHIC using a C API and binary arithmetic coding, whereas our implementation runs without such optimizations. Notably, MoRIC is implemented with PyTorch serial inference across all modules for research purposes, where substantial speedups are expected with appropriate engineering optimizations.
>
>   To improve clarity over latency, we have added the following clarification to the revised manuscript: "We want to note that all reported MoRIC results are based on unoptimized research code. Notably, our decoding modules are now executed serially during latency evaluation; a simple parallel execution can substantially reduce runtime. We also expect further improvements through advanced engineering optimizations such as C-API integration and binary arithmetic coding [2]."
>
>   [2] Blard, Théophile, et al. "Overfitted image coding at reduced complexity."
>
> - **W4.** Due to computational constraints, initial CLIC2020 results were provided in our supplementary materials (Table 1). We have since conducted comprehensive experiments with more RD values and adaptive architectures. (see **Table 1.1, 1.2, and 1.4** in our response to **[Ek8z]**.)
>
> **Response to each question:**
>
> - **Q1.** The reviewer is correct. We will revise the manuscript to include the following additional sentence: "C* coding approximates object contours using polygonal chains. Starting from an initial point, each contour segment is encoded by a fixed step length and a quantized offset angle relative to the previous direction. A move-to-front transform is applied to the angle sequence, alongside adaptive mechanisms such as global resets, offset adaptation, and contour dilation, to enhance compression efficiency." Experiments about the chain-coding are also provided in **Table 3.3** in our response to **[BiKZ]**.
>
> - **Q2.** The contour boundaries remain barely visible, even at low bitrates such as 0.05 bpp. This robustness stems from two key design choices in MoRIC:
>
>   (a). **Design of the global modulation network (GMN):** MoRIC integrates global-local context through its GMN and progressive modulation, effectively suppressing artifacts. Similar strategies have been shown to reduce artifacts in INR literature [3–4].
>
>   (b). **Region-wise fitting strategy:** Instead of fitting the image in a patch-wise manner, MoRIC performs region-wise fitting, enabled from C* coding, enhancing INR representation while reducing artifacts (as also observed in [3]).
>
>   Following the suggestion, we will add visualizations to show artifact suppression, especially at low BPPs. To further quantify this in the discussion period, we include an ablation study on GMN and region-wise fitting (exp-1 and exp-2) for artifacts analysis and quality assessments, reporting PSNR, MS-SSIM, PSNR-edge/PSNR-smooth edge (two metrics from [4] that assess artifacts via Canny edge detection with disk dilation). For exp-1, we compare MoRIC with a variant that removes the GMN component, denoted as MoRIC (w/o GMN).  Exp-2 replaces the target region of MoRIC with a rectangular patch to cover the same target, denoted as MoRIC (rect), where contour coding is omitted. Detailed performance with BD-rate increase for each variant of MoRIC is presented in **Table 4.1**.
>
> **Table 4.1-a  Ablation study for MoRIC (w/o GMN) for artifacts analysis on Kodak.**
>
> | **Rate**         | 0.1543  | 0.5399  | 0.7892  | 1.2003  | BD-rate | **Observed artifacts**         |
> |-|-|-|-|-|-|--|
> | **PSNR**   | 28.910  | 34.437  | 36.445  | 39.057  | +7.03| Ringing artifacts,             |
> | **MS-SSIM**      | 11.403  | 16.788  | 18.674  | 21.108  | +6.91   | Gradient discontinuity,        |
> | **PSNR_edge**    | 25.038  | 32.104  | 34.527  | 37.604  | +6.11 | Inconsistent textures          |
> | **PSNR_smooth**  | 31.960  | 35.628  | 37.326  | 39.635  | +8.56 |  |
>
> **Table 4.1-b Ablation study for MoRIC (rect) for artifacts analysis on Kodak dataset.**
>
> | **Rate**         | 0.1500  | 0.5399  | 0.7496  | 1.1157  | BD-rate | **Observed artifacts**         |
> |------------------|---------|---------|---------|---------|------------------------------|--------------------------------|
> | **PSNR**         | 29.067  | 34.509  | 36.645  | 39.317  | +2.53                        | Very visible                   |
> | **MS-SSIM**      | 11.518  | 16.911  | 18.857  | 21.273  | +2.28                        | Stitching artifacts            |
> | **PSNR_edge**    | 25.159  | 32.122  | 34.747  | 37.950  | +2.87                        | Between patches.               |
> | **PSNR_smooth**  | 32.171  | 35.720  | 37.490  | 39.817  | +2.02                        |                                |
>
> From Exp-1, we conclude the importance of GMN: without it, ringing artifacts and inconsistent texture scale can be observed. From Exp-2, we can observe noticeable stitching artifacts in MoRIC (rect) with reduced RD and perceptual performance, highlighting the effectiveness of the C* coding. To further assess MoRIC’s perceptual quality, we report MS-SSIM results under MS-SSIM-trained settings (**Table 4.2**). MoRIC achieves impressive performance, empowered by the above two design choices, closely matching advanced codecs like ELIC. Both **Table 4.1 and Table 4.2** results will be incorporated into the revised manuscript.
>
> [3] Ashkenazi, Maor, and Eran Treister. *"Towards croppable implicit neural representations."
>
> [4] Feng, Haoan, et al. *"SASNet: Spatially-Adaptive Sinusoidal Neural Networks."
>
> **Table 4.2: Detailed MS-SSIM / bpp performance on Kodak dataset.**
>
> | Codec     | Rate 1         | Rate 2         | Rate 3         | Rate 4         | Rate 5         | BD-rate vs. VTM (%) |
> |-|-|-|-|-|-|-|
> | VTM       | 11.963 / 0.1556 | 14.315 / 0.2871 | 16.751 / 0.4920 | 17.961 / 0.6276 | 20.303 / 0.9728 | 0|
> | MoRIC     | 13.910 / 0.1565 | 16.840 / 0.2782 | 19.607 / 0.4801 | 20.605 / 0.5953 | 22.802 / 0.8653 | -43.66 |
> | ELIC      | 12.229 / 0.0907 | 15.863 / 0.2154 | 18.830 / 0.3937 | -| 21.674 / 0.6669 | -46.94  |
> | MLIC++    | 13.299 / 0.0946 | 16.546 / 0.2115 | 18.177 / 0.3012 | 19.808 / 0.4372 | 21.432 / 0.6007 | -54.73 |
>
> - **Q3.** Following the reviewer’s suggestion, we conducted additional experiments by training MoRIC using masks with varying degrees of over- and under-segmentation errors.
>
>   Specifically, we simulate small and substantial segmentation errors via mask dilation and erosion, with IoU values of 14.22\%/34.18\% (under-segmentation) and 19.85\%/48.58\% (over-segmentation). Results are shown in **Table 4.3**.
>
>   We observe that MoRIC is robust to segmentation imperfections, especially when the segmentation IoU is below 20%, with performance degrading gradually as segmentation quality decreases. However, for both under- and over-segmentation, more significant performance drops begin to appear when the IoU approaches 30–40%. Additionally, substantial segmentation errors can introduce encoding instability, occasionally leading to suboptimal RD performance.
>
>   While MoRIC **does not currently optimize region selection**, we believe that better segmentation strategies could yield substantial improvements by leveraging this trade-off.
>
> **Table 4.3 Ablation study over segmentation error for MoRIC on Kodak dataset.**
>
> | Methods                     | IoU               | PSNR/bpp (1)         | PSNR/bpp (2)         | PSNR/bpp (3)         | PSNR/bpp (4)         | BD-rate |
> |-|-|-|-|-|-|-|
> | MoRIC (lossy C* coding)    | 3.95%             | 29.121 / 0.1516      | 34.599 / 0.5279      | 36.655 / 0.7449      | 41.259 / 1.4533      | 0       |
> | **Under-segmentation**     | 14.22%            | 29.035 / 0.1537      | 34.573 / 0.5293      | 36.570 / 0.7664      | 41.256 / 1.4609      | +2.11   |
> |                            | 34.18%            | 29.059 / 0.1530      | 34.569 / 0.5323      | 36.602 / 0.7604      | 41.042 / 1.5078      | +2.76   |
> | **Over-segmentation**      | 19.85%            | 29.066 / 0.1541      | 34.564 / 0.5327      | 36.624 / 0.7617      | 41.235 / 1.4666      | +2.09   |
> |                            | 48.58%            | 29.038 / 0.1553      | 34.562 / 0.5327      | 36.603 / 0.7591      | 41.064 / 1.5513      | +3.29   |

---

> > ### Comment · Reviewer_1oDR · 2025-08-05
> > **Official comment from reviewer 1oDR**
> >
> > Thank you for the detailed response. I appreciate the additional results on CLIC as well as the ablation study regarding the impact of segmentation errors on the compression performance. I will raise my score to "accept".

---

### Official Review · Reviewer_BiKZ · 2025-07-02

**Clarity:** 3
**Significance:** 3
**Originality:** 4
**Rating:** 5
**Confidence:** 4

**Summary:**

This paper introduced MoRIC, an image compression model based on implicit neural representations. MoRIC features region-based model fitting, i.e., instead of fitting a single model to every pixels, MoRIC assign different models to different regions, such that the models can better fit the local distributions. To achieve region-based coding, MoRIC extract the contours and apply chain coding, enabling coding with arbitrary shape regions, while maintaining high coding efficiency.

**Questions:**

- For evaluation with the single-object compression, do you compute the average over all pixels (including the masked pixels)?
- How are the baseline models being evaluated with the single object cases? Do you overfit the INR based codecs for the masked pixels as well?
- Is coding the contour more efficient than coding masks (with a high efficient codec)?
- Does the reported encoding/decoding time including everything, both of the entropy coding/decoding time/contour generation & compression?
- Why using DAVIS dataset? It isn’t a standard dataset for image compression. Also, in this dataset, the objects could only occupy very small part of the images, which the proposed method could have a larger advantage with the single object case.
- How were the ablation study models being built? Are the modules simply being removed? Did the models be adjusted for fixing the complexity & number of parameters?

**Ethical Concerns:**

["NO or VERY MINOR ethics concerns only"]

**Final Justification:**

A new image coding model is proposed in this paper with region/layered compression based on INRs. This is an important but yet to be explored topic. The codec performance is promising, although it performance is not groundbreaking in the general image compression task, but it supports layered coding which will could better serve certain applications. The evaluation and experiments are comprehensive, especially considering the additional details provided during rebuttal/discussion. The concerns raised have all been addressed. Overall, I think this is a technically solid paper.

**Limitations:**

yes.

**Quality:**

3

**Strengths And Weaknesses:**

Strengths
- The idea of applying region-based synthesis networks is novel and rational. In images, there are clearly different local distributions which fitting different synthesis networks can benefit the coding.
- Open sourcing the C* coding is an important contribution.
- The results of both single object/full image compression are promising.

Weaknesses
- The practical application of scalable layered compression has not been discussed. The proposed model still requires decoding the latent, plus computing the full shared components ($f_{\theta}$), even if the target is only part of the scene. The partially decoded image is not a low quality version of the original image, but an incorrect colour version of it. Also, progressively decode different regions also require caching the shared features/variables. The authors should consider discussing what is the advantage of decoding only part of the image (e.g. saving how many computation?).
- The region selection requires manually selecting, when the mask is not provided. This is not practical in most of the image compression applications.
- While the theoretical complexity is low, the performance/actual coding trade-off has not been evaluated. It is not clear if the improved performance worth the slow down of coding.

---

> ### Author Rebuttal · Authors · 2025-07-31
>
> We thank the reviewer for their valuable feedback. In response, we will:
> (a) include a discussion on practical applications for scalable layered compression;
> (b) discuss automatic partitioning alternatives;
> (c) add experiments analyzing the trade-off between performance and coding cost;
> (d) add experiments to validate the efficiency of C\* coding; and
> (e) include additional experiments on the CLIC2020 dataset.
>
> Next, we address the identified weakness and the questions point by point.
>
> - **W1.** To highlight scalable layered compression applications of MoRIC, we have revised the manuscript to include the following section:
>
>   "MoRIC supports progressive rate control with semantic consistency, enabling several practical applications:
>
>    (i) **Goal-oriented progressive compression**: Suited for scenarios like surveillance and edge AI, where full-resolution reconstruction is not always needed. Semantic outlines are sent first, with key regions refined on demand, saving bandwidth while preserving critical information.
>
>     (ii) **Distributed compression**: In multi-agent settings (e.g., robotics, NeRF), agents transmit only observed regions; others are inferred or selectively refined in a MoRIC fashion.
>
>   (iii) **Interactive content delivery (e.g., VR, AR, games)**: Decoding adapts online to user attention, delivering high-quality reconstruction in focus areas, while other regions are inferred and progressively refined as attention shifts.
>
>   (iv) **Video streaming/preview**: Coarse frames with accurate key regions and semantically consistent background are transmitted first, followed by progressive refinement for a seamless viewing experience."
>
>   To further illustrate the benefits of partial decoding, we report detailed **computation and bit cost savings** for a surveillance scenario in **Table 3.1**. For experimental details, we refer to Fig. 5 in our supplementary materials for a demo illustration.
>
>   MoRIC achieves up to 73.56% and 37.31% bitrate savings compared to VTM and MLIC++, respectively. It also provides substantial computational savings, thanks to its low decoding complexity, where the complexity of VVC is estimated to be comparable on standard hardware.
>
> **Table 3.1. Bitrate and computational savings enabled by partial decoding in MoRIC**
>
> | **Method** | **VTM** | **MLIC++** | **MoRIC** |
> |-----------|--------|-----------|----------|
> | **PSNR (dB)** | 52.301 | 52.332 | 52.584 |
> | **Bit cost (×10⁴)** | 2.107 | 1.667 | 1.214 |
> | **Bit saving vs. methods** | 73.56% | 37.31% | - |
> | **Computation saving** | 2× | 123× | - |
>
> - **W2.** We currently use SAM2 for region partitioning, which inherently supports automatic segmentation and can be extended with other advanced strategies such as saliency-based or prompt-guided segmentation for various compression scenarios. For alternative partition methods, we refer to our response to **reviewer [3Cij] (W1. and Q3)**. For robustness to segmentation quality, we refer to **Table 4.3**.
>
> - **W3.** In response to the comment, we include an experiment evaluating the performance–coding cost trade-off. We train MoRIC (lossy-fixed) and CoolChic-V4 on the Kodak dataset under encoding steps from 1k to 80k. The resulting BD-rates over VTM-19.1 are shown in **Table 3.2**. MoRIC achieves comparable RD performance with 2–3× fewer coding steps, attributed to its region-wise coding strategy, similar to the observations in [1,2].
>
>   Consequently, MoRIC can instead **potentially accelerate convergence** rather than slowing encoding. As shown in Table 3 and Fig. 3 in the supplementary, increasing the number of regions (from 1 to 3) can further improve convergence through more effective INR specialization.
>
>   Previously, encoding cost was typically addressed via engineering optimizations for an overfitted codec. In this context, a 2–3× algorithmic acceleration represents a meaningful step toward practical, low-latency compression. While our results reflect theoretical gains in convergence and complexity, actual latency also depends on deployment factors such as memory and hardware, where further improvements are expected.
>
>   [1] "Partition speeds up learning implicit neural representations based on exponential-increase hypothesis", 2023.
>   [2] Coordx: Accelerating implicit neural representation with a split MLP architecture, 2022.
>
>   In addition to gains in performance, acceleration, and convergence, MoRIC also enables applications such as efficient target-area compression and layered compression. Together, these results demonstrate the effectiveness of the MoRIC design.
>
> **Table 3.2. BD-rate over VTM-19.1 for different schemes across various encoding steps**
>
> | **Encoding steps** | 1,000  | 2,000  | 3,000  | 5,000  | 10,000 | 20,000 | 30,000 | 50,000 | 80,000 |
> |--------------------|--------|--------|--------|--------|--------|--------|--------|--------|--------|
> | **RD of MoRIC**    | 74.36% | 43.17% | 30.71% | 20.60% | 11.27% | 5.82%  | 3.61%  | 1.25%  | -0.91% |
> | **RD of CC-V4**    | 216.16%| 98.16% | 61.53% | 37.33% | 21.22% | 13.42% | 10.49% | 8.62%  | 6.59%  |
>
> **Answers to questions:**
>
> - **Q1.** For the evaluation of the single-object compression task, only the target region is considered, with masked pixels excluded from computation. We will clarify this in the revised manuscript.
>
> - **Q2.** We explored multiple design choices and adopted the best-performing configurations. By default, each baseline codec operates on the smallest rectangular region enclosing the target object, with masked pixels within this region jointly learned. For INR-based methods, we also evaluate variants combined with our proposed C\* coding algorithm for ablation studies on the architectural innovations. For more details of the baseline models for single-object cases, we refer to Section E.2 of the appendix (Lines 638–655).
>
> **Table 3.3. Contour coding efficiency on Kodak dataset (kodim23), evaluated by error rate / bit cost**
>
> |          Methods                      |   error rate  | /  bit cost     |        |
> |--------------------------------|--------|--------|--------|
> | **Lossless chain coding**      |0% / 1641 |
> | **Quality values of VTM**      | 10     | 36     | 56     |
> | **Performance of VTM**         | 0% / 9128 | 1.48% / 6864 | 6.62% / 1960 |
> | **Step length of C\***         | 5      | 10     | 15     |
> | **Performance of C\***         | 0.66% / 890 | 1.17% / 360 | 2.08% / 251 |
>
> - **Q3.** In response, we provide both theoretical analysis and experiments (**Table 3.3**) to support and verify our findings.
>
>   Whether contour coding is more efficient than mask-based encoding depends on the applications. When the goal is to only localize and compress object boundaries, chain coding can offer significant advantages (see **Table 3.3** as an example). Mask-based approaches typically require one bit per pixel across the entire image, whereas contour representations involve a significantly smaller set of pixels. Instead, a basic chain coding scheme can encode directional steps (e.g., 8 directions) using just 3 bits per step. More advanced methods that incorporate geometric cues and relative motion (such as VCC, 3OT, or NAD) can further enhance compression efficiency. These techniques are all included as open-source resources in our supplementary materials.
>
>   As shown in **Table 3.3**, our C\* coding method achieves lossless compression (error rate 0%) with a comparable bit cost to VTM’s mask coding (with an error rate of 6.62%). Moreover, by adjusting the step length, C\* coding can further reduce the bitrate with only minimal degradation in coding accuracy.
>
>   However, chain coding may be less efficient in scenarios with very noisy or highly irregular contours, where frequent direction changes and large curvature variations reduce coding efficiency. In such cases, mask-based approaches, especially those leveraging learned priors, can offer greater robustness.
>
>   In our setting, contour coding aligns naturally with the design of overfitted codecs and can be integrated seamlessly into the framework.
>
> - **Q4.** We have now reported a breakdown of the coding cost (**Table 2.1** of our response to **Reviewer [3Cij]**), including all components except range coding time, to ensure a fair comparison with C3. We will clarify this point in the revised manuscript.
>
> - **Q5.** We use the DAVIS dataset because it provides region-level ground truth, which aligns well with our experimental setup. Instead, standard compression datasets typically lack such annotations. Within our MoRIC framework, all region partitions are generated using SAM, and DAVIS allows us to validate our method on high-resolution data with reliable region labels. Additionally, we now include new experiments on the **CLIC2020 dataset** to further assess MoRIC’s performance (see **Tables 1.1, 1.2, and 1.4** in response to **Reviewer [Ek8z]**).
>
>   For small-object cases, baseline methods also benefit from reduced fitting regions (a smaller rectangular), while the relative parameter cost of the INR in MoRIC becomes more pronounced. Intuitively, MoRIC offers greater advantages in scenarios with complex object contours, where its explicit contour-based coding provides more substantial benefits.
>
> - **Q6.** For the ablation study in Table 7, modules are directly removed to assess their individual contributions, without adjusting for complexity or parameter count. We want to note that ensuring strict fairness in parameter count for overfitted codec is challenging, as additional parameters may introduce overhead and can sometimes even degrade RD performance. Here, we focus on validating the impact of the C coding scheme, PCM modulation, and LSN design, to assess whether introducing these mechanisms leads to improved RD performance.

---

> > ### Comment · Reviewer_BiKZ · 2025-08-05
> >
> > Thank you to the authors for their response. Here are my further questions:
> >
> > W1: I agree that scalable compression has its own applications. However, my original question concerns how the proposed method adapts to these scenarios. The computation savings reported in the reply do not reflect the benefit of partial decoding. For example, figure 7c shows that MoLIC already achieves over 150× computation savings comparing with MLIC++ even with full image decoding. Since partial decoding should yield additional savings, could you clarify how these savings in table 3.1 were calculated? Also, how did you obtain the complexity numbers for VTM?
> >
> > Q3: The mask should require at most one bit per pixel if we entropy-code it (which is also what the VTM doing in table 3.3). While the authors note that chain coding may be less efficient for complex shapes, the case studied here (Kodim23) is clearly a simple one. I suggest the authors include a more comprehensive evaluation of mask coding efficiency in the revised paper.
> >
> > Overall, I appreciate the authors’ detailed response—the additional information is indeed helpful. The response has addressed most of my concerns. While some experimental settings may not be ideal, I think that they are acceptable given the limitations of available datasets and the complexity of designing these experiments. I will take the authors’ response into account in my final justification.

---

> > > ### Author Response · Authors · 2025-08-06
> > >
> > > We thank the reviewer for the careful reading and for providing additional comments.
> > >
> > > **W1. (a):** In Table 3.1, there was a typo introduced during the formatting process: the reported computation saving for MLIC++ should be **1235x**  (not $123 \times$). We apologize for this oversight. The computation methodology is as follows: baseline methods compress the smallest rectangle covering the target region, which leads to extra computation on non-target areas. On average, this results in a 1.66x increase in computation in Table 3.1 (as the target region occupies around 60\% of the rectangle).
> > >
> > > The computation (MACs/Pixel) savings over MoRIC are computed as $(\text{Baseline} - \text{MoRIC})/\text{MoRIC}$:
> > > VTM: $(1000 \times 1.66 - 556)/556 \approx 2$,
> > > MLIC++: $(413935 \times 1.66 - 556)/556 \approx 1235$
> > >
> > > Similar coding saving trends appear in Fig. 18. Note that even baselines perform partial decoding here; full-image decoding would result in significantly higher cost.
> > >
> > > **W1. (b):** For VTM complexity in Table 3.1, we use the 1000 MACs/pixel at 0.15 bpp, which is reported in [1] as an average estimate  (Fig. 4 of [1]). (Since VVC does not use floating-point operations, this estimate is based on typical hardware equivalence.)
> > >
> > > [1] Ballé, Jona, et al. "Good, cheap, and fast: Overfitted image compression with Wasserstein distortion." CVPR. 2025.
> > >
> > > For clearer presentation, we will revise the manuscript with descriptions:
> > > “Thanks to the use of contour information, MoRIC achieves significantly lower decoding complexity in partial decoding scenarios by coding only the target region with simpler neural functions. As shown in Table 3.1, baseline methods require approximately 2x (VTM) and 1235x (MLIC++) more computation than MoRIC.”
> > >
> > > ---
> > >
> > > **Q3:** The reviewer is correct that a mask can require at most one bit per pixel after entropy coding. However, encoding the full mask can still be inefficient and expensive, especially when the goal is only to localize and compress object boundaries. This motivates our use of the proposed chain-coding method, which targets a more compact boundary representation.
> > >
> > > We agree with the reviewer that **including a case study of complex scenarios** would offer a more comprehensive evaluation of the chain-coding approach for boundary representation.
> > >
> > > Following the valuable suggestion, we include a much more complex version of the Kodim23 contour by adding random noise at each point of the boundary:
> > >
> > > $x = x + dx + \text{randint}(-3, 3), y = y + dy + \text{randint}(-3, 3)$.
> > > This results in a high-curvature contour, making the contour appear visually noisy and **sharply blurred** at each point.
> > >
> > > The comparisons are presented in **Table 3.4**. Compared to **Tables 3.3**, we observe that such complex contours require nearly $2\times$ the bit cost for lossless coding. VTM coding also becomes more complex, though still less efficient than chain coding; the gap narrows (e.g., at **Q=56**, it yields **lower bit cost but at the expense of accuracy**). We emphasize that this represents an extreme case, as such noisy contours are rare in practical scenarios.
> > >
> > > We highlight that our C* lossy coding is **less affected by contour noise**, as the incorporated dilation step of the C* algorithm acts as a denoiser, smoothing the contour slightly, thereby improving coding efficiency and reducing bit cost.
> > >
> > > **Table 3.4: Contour coding efficiency study for noisy contours on the Kodak dataset (Kodim23).**
> > > Each contour point is perturbed with random noise, and performance is evaluated in terms of error rate / bit cost.
> > >
> > > | **Methods**    | error rate / bit cost  |                 |                 |
> > > |------------------------------|-----------------|-----------------|-----------------|
> > > | **Lossless chain coding**    | 0% / 3664  |                 |                 |
> > > | **Quality values of VTM**    | 10              | 36              | 56              |
> > > | **Performance of VTM**       | 2.88% / 12192   | 4.6% / 8808     | 15.17% / 2128   |
> > > | **Step length of C\***       | 5               | 10              | 15              |
> > > | **Performance of C\***       | 1.60% / 1115    | 2.54% / 476     | 4.04% / 285     |
> > >
> > > We will include the above ablation study in the revised manuscript to enhance the comprehensiveness of the paper. We have carefully reviewed all other materials and sincerely thank the reviewer for the thoughtful and detailed questions. These comments have been very helpful, and we welcome any further suggestions or questions.

---

> > > > ### Comment · Reviewer_BiKZ · 2025-08-08
> > > >
> > > > Thank you the authors for the further response.
> > > >
> > > > This becomes make sense to me now, and I have no remaining concerns. I will change the rating accordingly.

---

### Official Review · Reviewer_3Cij · 2025-07-03

**Clarity:** 3
**Significance:** 2
**Originality:** 3
**Rating:** 4
**Confidence:** 2

**Summary:**

This paper introduces MoRIC, a novel image compression codec based on implicit neural representations (INRs). Unlike prior work that uses a single INR to model the entire image, MoRIC takes a modular, "divide and conquer" approach. It partitions an image into multiple, arbitrarily-shaped regions and assigns a dedicated, lightweight Local Synthesis Network (LSN) to compress each one. To ensure coherence across regions and prevent artifacts, a shared Global Modulation Network (GMN) injects global context into each LSN. A key component of this framework is a new adaptive chain coding method, named C* coding, designed to efficiently compress the region contours. The authors demonstrate that MoRIC achieves state-of-the-art rate-distortion performance for single-object and standard image compression among overfitted codecs, while maintaining low decoding complexity.

**Questions:**

Could you provide more detailed analysis on the trade-off between the number of regions (N) and overall complexity? Specifically, how does the bitrate for contours and network parameters scale with N? Is there an adaptive mechanism that could determine an optimal N for a given image to balance performance and overhead?

**Ethical Concerns:**

["NO or VERY MINOR ethics concerns only"]

**Final Justification:**

The authors' rebuttal has adequately addressed my concerns, and I am inclined to accept this paper.

**Limitations:**

yes

**Quality:**

3

**Strengths And Weaknesses:**

Strengths:
1.The proposed modular, region-based paradigm for overfitted INR codecs appears to be a novel contribution. This "divide and conquer" strategy is an intuitive way to potentially improve adaptation to local image statistics compared to monolithic models. This design could enable applications like fine-grained rate control and single-object compression.
2.The paper presents promising experimental results. According to the reported figures, MoRIC seems to achieve a strong rate-distortion performance, setting a new benchmark among overfitted codecs and even surpassing the classical VTM codec on the Kodak dataset.

Weaknesses:
1.A potential limitation is the method's reliance on manual or semi-manual region selection. While this is understandable for an initial study, it might pose a challenge for practical, fully automated systems. The performance of the codec could be sensitive to how these regions are defined.
2.The encoding process of the proposed method appears computationally intensive. The paper acknowledges this, and it's a known trade-off for this paradigm. This aspect may confine the method's applicability to specific scenarios where encoding time is not a critical constraint.

---

> ### Author Rebuttal · Authors · 2025-07-31
>
> We thank the reviewer for the valuable feedback. In response, we will: (a) include an ablation study on the robustness of partition strategy in MoRIC; (b) introduce an automatic partitioning algorithm with adaptive region count; (c) add experiments demonstrating MoRIC’s encoding advantages over other overfitted codecs; and (d) provide a detailed analysis of the trade-offs between region count, complexity, and bitrate.
>
> Next, we address the commented weakness and the question point by point.
>
> - **W1.** We thank the reviewer for highlighting this important point. While our current implementation uses SAM2 for region selection, we agree that fully automated partitioning is essential for practical deployment.
>
>   To this end, we note that MoRIC **supports automatic region partitioning** via connected component analysis on SAM2 outputs, enabling automatic selection of the top-k object regions. It is also compatible with task-adaptive strategies, including prompt-guided segmentation from SAM2, as well as alternatives like DINOv2 with DeepLabv3 or EML-Net for saliency-based partitioning, allowing enhanced adaptability and partition quality across diverse applications.
>
>   It is worth noting that the performance of MoRIC is **robust to segmentation imperfections**, thanks to its advanced PCM design and region-wise INR fitting, as demonstrated in our response to **Reviewer [1oDR] (Table 4.3)**.
>
>   We will add the following clarification to the revised manuscript:
>   *“MoRIC is robust to segmentation quality, supports automatic region partitioning via SAM2, and can incorporate task-adaptive strategies such as prompt-guided or saliency-aware segmentation to enhance partition quality and compression performance.”*
> - **W2.** High encoding complexity is a common phenomenon across **all overfitted coding paradigms** due to their instance-specific optimization, and is not unique to MoRIC. Potential solutions include meta-learning [1] and engineering optimizations [2], which have been shown to accelerate encoding by 10×–100×, reducing Kodak sample encoding time to around 20 seconds.
>
>   In this context, we would like to highlight another contribution of our work (previously discussed in Supplementary Section 3.1): **MoRIC mitigates encoding complexity to some extent** via a region-wise fitting strategy with shared modulation, which accelerates convergence and improves practical efficiency.
>
>   This design **significantly reduces** the number of required **encoding steps** compared to prior overfitted codecs, achieving a **2–3× algorithmic speedup** (see **Table 3.2** in our response to **Reviewer [BiKZ]**). These convergence gains, combined with potential engineering improvements, represent meaningful progress toward practical, low-latency overfitted codecs.
>
>   [1] Dupont, Emilien, et al. "COIN++: Neural Compression Across Modalities." *Transactions on Machine Learning Research*.
>
>   [2] Blard, Théophile, et al. "Overfitted image coding at reduced complexity." *2024 EUSIPCO*.
>
> **Answers to questions:**
>
> - **Q1.** Following reviewer's suggestion, we provide a detailed analysis of the trade-off between region count (N) and coding complexity. As shown in **Table 2.1**, the breakdown of coding cost is reported for various values of N (1–3). (Considering various existing implementations of entropy coding schemes and for fair comparison with [3], we exclude range coding time from our measurements, which is also a component of every classical codec.)
>
>     [3] Kim, Hyunjik, et al. "C3: High-performance and low-complexity neural compression from a single image or video." CVPR. 2024.
>
> **Table 2.1 Encoding (1000 steps) and decoding cost breakdown of MoRIC on Kodak dataset for different numbers of regions.**
>
> _SAM2 partition time: 533–830 ms (GPU); C* encoding: 119.58 ms (CPU); C* decoding: 12.86 ms (CPU)_
> | MoRIC (N=1 region) | **Module**| **ARM**| **GMN**| **LSN1**| **LSN2** | **LSN3** |
> |-|-|-|-|-|-|-|
> || GPU encoding time (s/1k steps)  | 7.382  (29.89%)| 12.331 (49.93%)   | 4.984 (20.18%) | – | – |
> || Encoding complexity kMACs/pixel   | 2175| 2589| 99| – | – |
> || CPU decoding time (ms) | 182.17 (53.03%)  | 153.35 (44.64%)   | 8.03 (2.34%)  | – | – |
> || Decoding complexity MACs/pixel    | 725 | 863 | 33 | – | – |
>
> | MoRIC (N=2 regions) | **Module**| **ARM**| **GMN**| **LSN1**| **LSN2**| **LSN3**|
> |-|-|-|-|-|-|-|
> || GPU encoding time (s/1k steps) | 7.279 (27.56%) | 13.298 (50.35%)| 2.878 (10.90%)| 2.956 (11.19%)|–|
> || Encoding complexity kMACs/pixel| 2175 | 3381| 99 | 99 |–|
> || CPU decoding time (ms)| 182.85 (44.71%)  | 209.45 (51.21%)   | 8.05 (1.97%) | 8.62 (2.11%)|–|
> || Decoding complexity MACs/pixel|725|1127| 33| 33|–|
>
> | MoRIC (N=3 regions) | **Module** | **ARM**| **GMN**|**LSN1**|**LSN2**|**LSN3**|
> |-|-|-|-|-|-|-|
> || GPU encoding time (s/1k steps) | 6.961 (21.71%)   | 15.512 (48.38%)| 2.960 (9.23%) | 2.703 (8.43%) | 3.926 (12.25%)|
> || Encoding complexity kMACs/pixel  | 2175| 4173  | 99 | 99  | 99 |
> || CPU decoding time (ms)| 185.28 (38.09%)  | 275.57 (56.65%) | 8.39 (1.72%)| 8.10 (1.67%) | 9.14 (1.88%)|
> || Decoding complexity MACs/pixel   | 725| 1391 | 33 | 33 | 33 |
>
> From the table, we observe that increasing the number of regions leads only to a slight rise in overall complexity. Specifically, each additional local INR adds 33 MACs/pixel from LSNs and 264 MACs/pixel from GMN in the forward pass. The added cost primarily arises from extra operations in the shared modulator to accommodate more local INRs (see Fig. 5 in our manuscript).
>
> We also recognize the importance of real-world latency and thereby include a breakdown of actual coding time across regions to provide practical insights. Here the time for the partition and contour coding is also provided. From our experiments, ARM exhibits consistent time costs across different region settings. Each additional LSN adds approximately $8$–$9$ ms to decoding time and $2$–$4$ seconds for 1000 encoding steps. A similar trend is observed for GMN, which dominates the total coding time and increases significantly as the number of regions grows, consistent with its complexity analysis. Note that all reported results are based on the same unoptimized code, with decoding executed serially. Significant speedups are expected for MoRIC with proper engineering optimizations.
>
> Overall, MoRIC demonstrates strong scalability with respect to region count, as the added complexity and latency remain acceptable relative to its strong rate-distortion performance. We will include the above discussion and **Table 2.1** in the revised manuscript to provide a more comprehensive analysis of complexity and coding time across different region numbers.
> - **Q2.** We repeat the experiments from Fig. 9 in the manuscript and provide a detailed breakdown of the bitrate share between contours, network parameters, and latent for different rate regimes and region numbers $N$ (see **Table 2.2**).
>
>   From the table, we can observe that C* coding can compress contours with minimal bitrate overhead, remaining as low as 1.89% for $N=2$ and increasing only slightly with more regions (up to 3.20% for $N=3$). As bitrate increases, the relative impact of contour cost becomes negligible, allowing MoRIC to achieve competitive RD performance with finer region partitioning efficiently.
>
>   We will include the above discussion and **Table 2.2** in the revised manuscript to provide a more comprehensive analysis of bitrate distribution across different compression rates and region numbers.
>
> **Table 2.2 Bit rate breakdown for MoRIC across region numbers on Kodak dataset.**
>
> | **Bit cost share**| **Low bpp** | **Medium bpp** | **High bpp** | **BD-rate vs. MoRIC (single)** |
> |-|-|-|-|-|
> | **MoRIC (single region)**  | 0.11815 | 0.41885  | 1.27234 | 0 |
> | Contour/Parameters/Latent (%) | 0% / 25.39% / 74.61% | 0% / 8.14% / 91.86% | 0% / 2.89% / 97.11% |                                |
> | **MoRIC (N=2 regions)**  | 0.11940 | 0.41879 | 1.24859 | -1.29% |
> | Contour/Parameters/Latent (%) | 1.89% / 26.01% / 72.09% | 0.54% / 8.41% / 91.05% | 0.18% / 3.11% / 96.71% |                                |
> | **MoRIC (N=3 regions)**  | 0.12067| 0.41567| 1.24135| -1.72%|
> | Contour/Parameters/Latent (%) | 3.20% / 26.71% / 70.08% | 0.93% / 8.55% / 90.52% | 0.31% / 3.22% / 96.47% |                                |
> - **Q3.** There exist adaptive strategies for determining the number of regions and corresponding partitions for MoRIC. In general, images with substantial regional variation benefit from using more local INRs, which enables faster fitting and improved compression performance.
>
>   This observation suggests that the problem of getting the adaptive region number can be transformed into an analysis of the complexity of an image’s regional distribution. While a full investigation of how region distribution variation impacts partitioning and corresponding RD performance is beyond the scope of this paper, we consider this a valuable suggestion and a promising direction for future development of implicit codecs. To this end, we outline two practical solutions:
>
>   (a) **Feature-driven region partitioning**: Leverage a strong backbone such as DINOv2 for robust region-feature extraction and semantic grouping, followed by SAM2 to perform region segmentation guided by these cues. This combination enables adaptive region partitioning that captures region-wise semantic structure, where more complex images naturally result in more regions guided by semantic grouping.
>
>   (b) **Saliency-guided region partitioning**: A learning-based saliency model such as EML-Net can be used to guide partitioning by grouping visually important or semantically similar areas. Subsequent posterior analysis and filtering can further refine the selection to balance coding complexity and rate-distortion performance.
>
>   These approaches demonstrate MoRIC’s versatility and strong potential for integration with emerging methods and diverse application scenarios.

---

> > ### Comment · Reviewer_3Cij · 2025-08-06
> >
> > Thank you for your comprehensive response, which has addressed my concerns. I will maintain my score.

---

> ### Comment · Area_Chair_aRij · 2025-08-06
>
> Dear reviewer,
>
> Thank you for the review.
>
> As the discussion phase is ending soon, can you please go over the authors' response and acknowledge it?
>
> Best,
> AC

---

### Official Review · Reviewer_Ek8z · 2025-07-04

**Clarity:** 2
**Significance:** 3
**Originality:** 3
**Rating:** 3
**Confidence:** 3

**Summary:**

The proposed MoRIC is a novel INR-based image compression method that represents each region with corresponding shape and texture codes, improving local adaptation and enabling fine-grained rate-distortion control. MoRIC achieves state-of-the-art single-object compression with lower decoding complexity, holds promises for fixed-background scenarios.

**Questions:**

1. Provide detailed descriptions of the encoding time.
2. Provide the performance of commercial compressors.
3/ The necessity of including the results on general natural scenes, since the proposed approach does not have clear advantageous on such data.

**Ethical Concerns:**

["NO or VERY MINOR ethics concerns only"]

**Limitations:**

yes

**Paper Formatting Concerns:**

looks good, but I this a more detailed figure caption will be helpful.

**Quality:**

3

**Strengths And Weaknesses:**

Strength
1. The rationale behind the proposed approach is inspiring and fits the fixed-background applications well.
2. The idea of progressive encoding is an attractive feature for INR-based compression, which permits saliency-specific compression.

Weakness
The manuscript lacks descriptions and discussions of the cons of the proposed approach, such as
1. What are the quality, expenses, and criteria of region partitioning? The partitioning might result in non-reasonable over-segmentation / under-segmentation, and the time consumption in partitioning should also be counted in the compression running time. The partitioning can be from a semantic or pixel perspective, which one is favoured?
2. I expect quantitative figures on the encoding time and the time spent on different modules.
3. The performance on general natural scenes is not that promising (Fig. 7); the decompression is faster than four competitors. can you explain why? Is it attributed to the region-specific strategy?
4. I understand that the method is INR-based and might be inferior to conventional commercial coding, such as H. 266, I still expect a comparison with the sota compressors, which is important to evaluate the performance of the proposed approach.

---

> ### Author Rebuttal · Authors · 2025-07-31
>
> We thank the reviewer for the valuable feedback. In response, we will: (a) add descriptions of our partitioning strategy and include ablation studies on the impact of segmentation quality; (b) provide a detailed breakdown of coding cost across various regions, along with corresponding theoretical complexity analysis; (c) include additional experiments on general natural scenes (CLIC2020 dataset); and (d) incorporate more baseline comparisons in the revised manuscript.
>
> Next, we address the commented weakness and the questions point by point.
>
> - **W1.** The current MoRIC region partitioning is guided by semantics and based on SAM2 [1]. The reviewer is correct that partitioning may result in unreasonable segmentation. However, MoRIC **does not rely on highly accurate segmentation**, thanks to the robustness of its GMN design and INR fitting capability. In fact, while lossy contour coding may cause minor over-segmentation, the resulting bit savings without significantly sacrificing distortion often lead to slightly improved rate-distortion trade-offs.
>
>     To evaluate this, we conducted experiments assessing the impact of segmentation quality under both over- and under-segmentation scenarios. The results are presented in **Table 4.3** (in our response to **Reviewer [1oDR]**).
>
>     The partitioning time of MoRIC is around 533-830 ms (3090GPU) for Kodak samples, which is reported in **Table 2.1** (in our response to **Reviewer [3Cij]**) as part of the coding cost. Since overfitted codecs are typically encoded offline, this one-time partitioning will have minimal impact on practical deployment.
>
>     In summary, MoRIC favors semantic partitioning over pixel-level precision due to its robustness to contour errors. Experimental results show that this design is both efficient and effective, while also maintaining strong perceptual quality, as demonstrated by the MS-SSIM experiments in **Table 4.2** (in our response to **Reviewer [1oDR]**).
>
>     [1] Ravi, Nikhila, et al. "SAM 2: Segment Anything in Images and Videos." ICLR 2025.
> - **W2.** In addition to Table 4 in the manuscript, we will include a module-wise breakdown of coding time and a theoretical complexity analysis (which is detailed in **Table 2.1** from our response to **Reviewer [3Cij]** due to space limitations) into our manuscript. Specifically, the backward pass complexity is approximated to twice the MACs of the forward pass, following [2].
>
>     [2] Baydin, Atilim Gunes, et al. "Automatic differentiation in machine learning: a survey." *Journal of Machine Learning Research* (2018).
> - **W3. (a) Performance concerns**: We would like to highlight that MoRIC sets a **new state-of-the-art** among overfitted codecs, **outperforming VTM (H.266)** on natural scene datasets such as **Kodak and CLIC2020**, a strength also noted by Reviewers [3Cij], [BiKZ] and [1oDR]. While MoRIC falls short of leading neural codecs like MLIC++ in RD performance, it achieves over 100× lower decoding complexity.
>
>     To further demonstrate its advantages on general natural scenes, we conducted additional experiments on the **CLIC2020 dataset** (as shown in **Tables 1.1, 1.2, and 1.4**), where even more pronounced gains were observed (**approaching MLIC and outperforming ELIC on CLIC2020**). Additionally, we also provide MS-SSIM results for MoRIC (see **Table 4.2** in our response to **Reviewer [1oDR]**), where **MoRIC approaches ELIC on Kodak**. Taken together, these results offer a comprehensive evaluation and underscore the promise of MoRIC, with evidence that is both strong and well-supported.
>
>     Beyond the RD experimental gains, MoRIC introduces broader contributions, including a modulated local-region fitting strategy, a chain-coding paradigm, and faster encoding convergence (see **Table 3.2** in our response to **Reviewer [BiKZ]**). These advances offer independent value for both INR and overfitted codec design. We welcome further suggestions to strengthen the soundness and impact of this work.
>
> **Table 1.1 Detailed RD datapoints of MoRIC on CLIC2020 (professional validation dataset).**
>
> | **Rate (bpp)**   | 0.0852 | 0.1353 | 0.3533 | 0.5140 | 0.7898 | 1.0721 |
> |------------------|--------|--------|--------|--------|--------|--------|
> | **PSNR (dB)**    | 30.6281 | 32.1680 | 35.9424 | 37.5443 | 39.7148 | 41.3063 |
>
> **Table 1.2 Comparison of different schemes for standard full image compression on CLIC2020.**
>
> | **Methods** | MLIC++ | MLIC | CST | EVC (SS/MM/LL) | C3 (fixed/adapt.) | **MoRIC (ours)** |
> |-------------|--------|------|-----|----------------|-------------------|------------------|
> | **BD-rate over VTM-19.1  (%).** | -16.31 | -12.12 | 7.4 | 7.93 / 1.45 / -0.80          | -3.61 / -6.47 |  -3.14 / -6.37 / -8.22 / **-10.58** |
> | **Decoding complexity (MACs/pixel)** | 414k | 447k | 583k | 35k / 124k / 258k     | 2925 |  1378 / 1659 / 1844 / **2882** |
> - **W3. (b) Decoding speed**: MoRIC achieves the **fastest CPU decoding speed** among all methods in Table 4 (object compression) of the manuscript. Specifically, most of the computation in overfitted codecs lies in the entropy model and synthesis module. MoRIC addresses this by two key design choices:
>
>   - **Explicit contour representation:** MoRIC skips the expensive auto-regressive decoding for latents outside target regions.
>   - **Progressive Concatenated Modulation (PCM):** This design enables efficient region-wise reconstruction under lightweight architecture. As a result, the synthesis module requires just **262 MACs/pixel**, which is approximately **81%** of COOL-CHICv4 and **62%** of C3, while still delivering superior RD performance.
>
>     To improve clarity, we will incorporate the paragraphs above, the coding cost analysis (**Table 2.1** (in our response to **Reviewer [3Cij]**)), and the CLIC2020 experiment results (**Table 1.1, 1.2, and 1.4**) into the revised manuscript.
> - **W4.** We would also like to respectfully clarify a possible misunderstanding: we have included both **VTM-19.1** (the reference implementation of **H.266**) and **MLIC++** (current state-of-the-art neural codec) for comparison in our manuscript. And as shown in our experiments, MoRIC **outperforms VTM-19.1 (H.266)** on standard image compression benchmarks such as **Kodak and CLIC2020**. Additionally, in single-object compression, MoRIC **surpasses** MLIC++ while **achieving 100×–1000× lower decoding complexity**. This advantage is particularly significant in high-bpp regimes.
>
>    To address the reviewer’s concern, we will add more baselines (**Table 1.3 and 1.4**) and revise the manuscript to more clearly highlight the baseline choices, as follows:
>
> **Table 1.3 More baselines on Kodak dataset.**
>
> | **Traditional/INR codec** | **JEPG2000** | **HEVC (HM)** | **BPG** | **MSCN** | **VC-INR** | **RECOMBINER** |
> |---------------------------|--------------|---------------|---------|----------|------------|----------------|
> | BD-rate over VTM-19.1     | 95.42%       | 24.90%        | 22.53%  | 115.91%  | 73.71%     | 63.86%         |
>
> | **Autoencoder codec**     | **BMS**      | **MBT**       | **CST** | **WYH**  | **STF**    | **ELIC**       |
> |---------------------------|--------------|---------------|---------|----------|------------|----------------|
> | BD-rate over VTM-19.1     | 30.34%       | 10.85%        | 3.99%   | -0.44%   | -3.96%     | -6.85%         |
>
> **Table 1.4 More baselines on CLIC2020 dataset.**
>
> | **Image codec**     | **BPG** | **ELIC** | **Cheng2020** | **WYH** | **STF** | **CCv4** | **MoRIC (ours)** |
> |---------------------|--------|----------|----------------|--------|--------|--------|------------------|
> | BD-rate over VTM-19.1 | 39.28% | -7.04%   | 7.22%          | -4.77% | -3.22% | -6.58% | -10.58%          |
>
>   “We evaluate MoRIC on the DAVIS, Kodak, and CLIC2020 datasets, comparing it with classical codecs (VTM, the reference implementation of H.266), autoencoder-based neural codecs (EVC for low complexity; MLIC++ for state-of-the-art RD performance), and recent overfitted codecs (C3 and COOL-CHIC v4). Additional baselines are provided in the appendix.”
>
>    “MoRIC surpasses state-of-the-art neural codecs in single-object compression, and outperforms VTM on standard compression benchmarks. These results highlight progress on key challenges in overfitted codecs, including achieving high RD performance with low decoding complexity and faster encoding through region-wise coding.”
>
> **Answers to questions:**
>
> - **Q1.** In our response to W2, we provided a detailed breakdown of both the theoretical and practical coding costs (presented as **Table 2.1** in our response to Reviewer [3Cij]).
>
> - **Q2.** In our response to W4, we incorporated additional baselines (shown in **Tables 1.3 and 1.4**).
>
> - **Q3.** We added experiments on the **CLIC2020** dataset (a standard benchmark with 41 high-resolution images) and included additional MS-SSIM results on Kodak, demonstrating promising performance gains (see **Tables 1.1, 1.2, 1.4, and 4.2**).

---

> > ### Author Response · Authors · 2025-08-08
> >
> > Dear reviewer,
> >
> >    As the discussion phase is coming to a close, we’d like to follow up to check whether our replies have addressed the concerns.
> >
> >    We sincerely thank reviewers for their time and thoughtful comments, which have been very helpful in improving our work. Please feel free to let us know if you have any further questions.
> >
> > Best,
> >
> > Authors

---

> ### Comment · Area_Chair_aRij · 2025-08-06
>
> Dear reviewer,
>
> Thank you for the review.
>
> As the discussion phase is ending soon, can you please go over the authors' response and acknowledge it?
>
> Best,
> AC

---

### Comment · Area_Chair_aRij · 2025-08-06

Dear reviewers and authors,

The discussion period is ending soon, so please make sure to finalize the discussion (and make mandatory acknowledgments, if not completed yet). Thank you so much for your active participation!

Best,
AC.

---

### Decision · Program_Chairs · 2025-09-17

**Decision:**

Accept (poster)

**Comment:**

This paper introduces MoRIC, an implicit neural image codec which partitions an image into regions and compress each region with local synthesis networks. Key technical components are the global modulator and the contour coding scheme. Overall, the reviewers agreed that the proposed framework is new and reasonable approach that achieves strong empirical results. There has been some concerns, including the limited scope of evaluation (DAVIS and Kodak), high encoding cost, and semi-manual nature of the algorithm. During the rebuttal period, most concerns have been addressed carefully by the authors (including experiments on CLIC), which are mostly acknowledged by the reviewers. I recommend acceptance.